# A Comprehensively Tight Analysis of Gradient Descent for PCA

**Zhiqiang Xu, Ping Li**

Cognitive Computing Lab
Baidu Research
No. 10 Xibeiwang East Road, Beijing 100193, China
10900 NE 8th St. Bellevue, Washington 98004, USA
{xuzhiqiang04,liping11}@baidu.com

## Abstract

We study the Riemannian gradient method for PCA on which a crucial fact is that despite the simplicity of the considered setting, i.e., deterministic version of Krasulina's method, the convergence rate has not been well-understood yet. In this work, we provide a general tight analysis for the gap-dependent rate at $O(\frac{1}{\Delta} \log \frac{1}{\epsilon})$ that holds for any real symmetric matrix. More importantly, when the gap $\Delta$ is significantly smaller than the target accuracy $\epsilon$ on the objective sub-optimality of the final solution, the rate of this type is actually not tight any more, which calls for a worst-case rate. We further give the first worst-case analysis that achieves a rate of convergence at $O(\frac{1}{\epsilon} \log \frac{1}{\epsilon})$. The two analyses naturally roll out a comprehensively tight convergence rate at $O(\frac{1}{\max\{\Delta,\epsilon\}} \log \frac{1}{\epsilon})$. Particularly, our gap-dependent analysis suggests a new promising learning rate for stochastic variance reduced PCA algorithms. Experiments are conducted to confirm our findings as well.

## 1  Introduction

Gradient descent is a basic, well-established, and celebrated optimization method for minimizing convex functions [11, 3], but remains far less understood theoretically for non-convex functions though it has been proven useful in practice for many non-convex problems [22, 17]. In the past decade, the research on gradient methods for non-convex problems has been gaining increasing attention in optimization and especially machine learning [6, 8, 4]. In this work, we focus on a notable example, namely the Riemannian gradient descent for the leading eigenvector computation of a real symmetric matrix $\mathbf{A} \in \mathbb{R}^{n \times n}$, which lays the foundation of Krasulina's method for PCA. This problem can be solved by either a traditional projection based method from the numerical algebra, such as the well-known power method or Lanczos algorithm [13], or a gradient search based method for the associated optimization problem such as the Riemannian gradient descent [18, 21]. The projected gradient descent (with constant step-sizes), as another closely related solver, can be interpreted either as a projection method or as a search method in this case (see Section 6 for interpretations). Compared to projection methods which have arguably been well-understood already [13, 9, 10], the search method, i.e., the Riemannian gradient descent, is lacking a clear and deep theoretical understanding on their convergence behaviors. This may largely be because the underlying optimization problem is geodesically non-convex:

$$\min_{\mathbf{x} \in \mathbb{S}^{n-1}} f(\mathbf{x}) = -\frac{1}{2}\mathbf{x}^\top \mathbf{A}\mathbf{x}, \tag{1}$$

where the objective function $f(\mathbf{x})$ is constrained on the $(n-1)$-dimensional unit sphere manifold $\mathbb{S}^{n-1} = \{\mathbf{x} \in \mathbb{R}^{n \times 1} : \|\mathbf{x}\|_2 = 1\}$ [1].

35th Conference on Neural Information Processing Systems (NeurIPS 2021).

Previous analyses of the Riemannian gradient descent for Problem (1) achieve a rate of (global[1]) convergence at $O(\frac{1}{\Delta^2} \log \frac{1}{\epsilon})$ which depends quadratically on the relative gap $\Delta$ between $\mathbf{A}$'s two (distinct) largest eigenvalues [20, 19][2]. The poor dependence of the rate on the gap characterizes a loose upper bound on the iteration complexity of the method, due to the artifacts of the used analysis techniques. Ding et al. [5] gave a tight convergence rate at $O(\frac{1}{\Delta} \log \frac{1}{\epsilon})$. However, the analysis is limited to positive definite matrices. This means that the general tight convergence analysis for real symmetric matrices, in fact, has not been settled. Our *first goal* in this work is to give a general analysis for the tight rate of this type, i.e., dependent on the gap, which is termed as the (tight) gap-dependent analysis hereafter. More importantly, when the gap is significantly smaller than the target accuracy $\epsilon$ on the objective sub-optimality of the final solution rather than its point distance to the optimal solutions, the rate of this type is actually not tight any more, which calls for a worst-case rate independent of any gap. Our *second goal* is to further give the first worst-case or gap-independent analysis being able to achieve a rate of convergence at $O(\frac{1}{\epsilon} \log \frac{1}{\epsilon})$. The two analyses naturally roll out a comprehensively tight rate of convergence at $O(\frac{1}{\max\{\Delta,\epsilon\}} \log \frac{1}{\epsilon})$. Our analyses are based on novel adaptations of the analyses for projection methods working for constant matrices [7, 10] to varying matrices during iterations. We also make a comparison between projection and search methods to help understand their subtle differences. Particularly, the gap-dependent analysis suggests a new step-size for search methods which gives rise to a novel learning rate for stochastic variance reduced PCA algorithms. Experiments verify our theoretical findings and demonstrate the effectiveness of the new learning rate. The main contributions are summarized as follows:

- We prove the first comprehensively tight convergence rate of the Riemannian gradient descent for Problem (1) at $O(\frac{1}{\max\{\Delta,\epsilon\}} \log \frac{1}{\epsilon})$.

- We propose an adaptive learning rate scheme for stochastic variance reduced PCA algorithms which can dramatically improve their convergence.

- We experimentally verify the established theory and demonstrate the effectiveness of the proposed adaptive learning rate.

## 2  Related Work

We focus on the gradient methods for the leading eigenvector computation. The classic Oja's algorithm and Krasulina's method, as stochastic counterparts of the projected gradient descent and Riemannian gradient descent with diminishing step-sizes, respectively, converge at a rate $O(\frac{1}{\Delta^2 \epsilon})$ [2] (Theorem 1.1 where $\mathbb{E}[\Psi_T] = O(\frac{c^2}{T})$ and $c = O(\frac{1}{\Delta})$ using their notations except for the iteration number $T$). The quadratic dependence on the gap was originally omitted there, probably because the gap was regarded as a constant. However, as argued in Musco et al. [10], this is insufficient. Interestingly, there are much fewer theoretical studies of the two methods in the deterministic setting. Absil et al. [1] introduced the Riemannian gradient descent for Problem (1) (Algorithm 2 there) with the Armijo line search for step-sizes, and implicitly stated a rate at $O(\frac{1}{\Delta} \log \frac{1}{\epsilon})$ (Theorem 4.6.3-iii) which, however, is local (Theorem 4.6.3-ii). Wen et al. [17] proposed a special retraction for the Riemannian gradient descent based on the Cayley transform (Algorithm 1). It was empirically found comparable and more stable than the Lanczos algorithm, however, only with the guarantee of convergence to a critical point (Theorem 2). Since Problem (1) is geodesically non-convex, theories established for geodesically convex functions [23] can't be applied unless locally. Note that we only consider global convergence in this work. Xu et al. [19] specifically analyzed the Riemannian gradient descent for Problem (1), and originally stated a rate $O\left(\min\{\frac{1}{\Delta^2} \log \frac{1}{\epsilon}, \frac{1}{\epsilon}\}\right)$ (Theorem 1). However, the constituent part $O(\frac{1}{\epsilon})$ resulted from the same mistake[3] as made in Balsubramani et al. [2] and mentioned at the beginning of this section. This means that their analyses are all gap-dependent and the better one of the rates is $O(\frac{1}{\Delta^2} \log \frac{1}{\epsilon})$. It is also the case for the work of Xu and Gao 2018 [20] (Theorem 4.1). Particularly, Ding et al. [5] improved the dependence of these rates on the gap from quadratic to linear at $O(\frac{1}{\Delta} \log \frac{1}{\epsilon})$. As explained in the previous section, however, it is not general. In

---

[1]Only global convergence is considered throughout the paper.

[2]It needs be noted there that the rate $O(\frac{1}{\epsilon})$ was mistakenly stated and actually should be $O(\frac{1}{\Delta^2 \epsilon})$ according to their analyses. See the next section for explanations.

[3]Precisely, $\psi(\mathbf{x}_T) = O(\frac{c^2}{T})$, where $c = O(\frac{1}{\Delta_p})$ implied by $a$'s expression, using their notations.

Table 1: Comparison of convergence rates of the Riemannian gradient descent for Problem (1).

|  | RATE | A | TIGHT | GAP-FREE COVERED |
|---|---|---|---|---|
| XU ET AL. [19] | $O(\frac{1}{\Delta^2} \log \frac{1}{\epsilon})$ | REAL SYMMETRIC | NO | NO |
| DING ET AL. [5] | $O(\frac{1}{\Delta} \log \frac{1}{\epsilon})$ | POSITIVE DEFINITE | YES | NO |
| THIS WORK | $O(\frac{1}{\max\{\Delta,\epsilon\}} \log \frac{1}{\epsilon})$ | REAL SYMMETRIC | YES | YES |

addition, to the best of our knowledge, before our work, there has been no worst-case or gap-free analysis of the Riemannian gradient descent for Problem (1). Comparisons are summarized in Table 1.

Shamir [14] proposed a learning rate scheme for stochastic variance reduced PCA algorithm, namely VR-PCA, based on Oja's algorithm. It can be used for VR-PCA's variant as well that is based on Krasulina's method [2, 16]. In this work, we put forward yet another learning rate scheme which we find much more effective than that suggested by [14] for these stochastic variance reduced PCA algorithms.

## 3   Notions and Notations

The only assumption on the given matrix $\mathbf{A}$, made in our analyses, is real symmetry, i.e., $\mathbf{A}^\top = \mathbf{A} \in \mathbb{R}^{n \times n}$. Let $\mathbf{A}$'s orthonormal eigenvectors be $\mathbf{v}_1, \mathbf{v}_2, \cdots, \mathbf{v}_n \in \mathbb{S}^{n-1}$, corresponding to eigenvalues $\lambda_1, \lambda_2, \cdots, \lambda_n \in \mathbb{R}$, i.e., $\mathbf{A}\mathbf{v}_i = \lambda_i \mathbf{v}_i$, in descending order. If $\lambda_1 > \lambda_n$, then let $\Delta_p$ $(1 \le p \le n-1)$ be the first nonzero eigenvalue gap, i.e., $\Delta_p = \lambda_p - \lambda_{p+1} > 0$, such that $\lambda_1 = \cdots = \lambda_p > \lambda_{p+1}$. Particularly, the first nonzero relative eigenvalue gap is defined as $\Delta = \frac{\Delta_p}{\lambda_1 - \lambda_n}$ if $\lambda_1 > \lambda_n$, otherwise as $\Delta = +\infty$. The update equations of the two gradient methods for Problem (1) with constant step-sizes are as follows:

$$\text{Projected gradient descent (PGD)}: \qquad \mathbf{x}_{t+1} = \frac{\mathbf{x}_t - \eta \nabla f(\mathbf{x}_t)}{\|\mathbf{x}_t - \eta \nabla f(\mathbf{x}_t)\|_2}, \qquad (2)$$

$$\text{Riemannian gradient descent (RGD)}: \qquad \mathbf{x}_{t+1} = \frac{\mathbf{x}_t - \eta \widetilde{\nabla} f(\mathbf{x}_t)}{\|\mathbf{x}_t - \eta \widetilde{\nabla} f(\mathbf{x}_t)\|_2}, \qquad (3)$$

where $\nabla f(\mathbf{x}) = -\mathbf{A}\mathbf{x}$ and $\widetilde{\nabla} f(\mathbf{x}) = -(\mathbf{I} - \mathbf{x}\mathbf{x}^\top)\mathbf{A}\mathbf{x}$ (throughout the paper $\mathbf{I}$ stands for an appropriately sized identity matrix) represent the Euclidean and Riemannian gradients of $f(\mathbf{x})$, respectively, $\eta > 0$ constant step-size, and denominators for normalization, i.e., $\mathbf{x}_{t+1} \in \mathbb{S}^{n-1}$.

Consider **Algorithm** that the Riemannian gradient descent for Problem (1) with constant step-size $\eta$ starts from a random initial point $\mathbf{x}_0 = \frac{\widehat{\mathbf{x}}_0}{\|\widehat{\mathbf{x}}_0\|_2} \in \mathbb{S}^{n-1}$ (where $\widehat{\mathbf{x}}_0$ is entry-wise standard Gaussian), and then repeatedly runs Equation (3) for $T$ iterations. We will focus on the analysis of this algorithm next. The results for the projected gradient descent are immediate by a much simpler path through the same analysis.

## 4   Gap-Dependent Analysis

Let $\mathbf{V}_p = [\mathbf{v}_1 \cdots \mathbf{v}_p]$. The solution space to Problem (1) then is $\{\mathbf{V}_p \boldsymbol{\alpha} : \boldsymbol{\alpha} \in \mathbb{S}^{p-1}\} \triangleq \mathcal{V}$, where $\mathbb{S}^0 = \{\pm 1\}$. To study the convergence of iterates to $\mathcal{V}$, it suffices to analyze the principal angle between $\mathbf{x}_t$ and $\mathbf{V}_p$, i.e., $\theta(\mathbf{x}_t, \mathbf{V}_p) = \cos^{-1}(\|\mathbf{V}_p^\top \mathbf{x}_t\|_2)$. Let $\mathbf{V}_{-p} = [\mathbf{v}_{p+1} \cdots \mathbf{v}_n]$ and $\square^\dagger$ represent the pseudo-inverse of a matrix. Then $\tan \theta(\mathbf{x}_0, \mathbf{V}_p) = \|(\mathbf{V}_{-p}^\top \widehat{\mathbf{x}}_0)(\mathbf{V}_p^\top \widehat{\mathbf{x}}_0)^\dagger\|_2$. Suppose that $T$ iterations are sufficient for $\sin^2 \theta(\mathbf{x}_t, \mathbf{V}_p) = 1 - \|\mathbf{V}_p^\top \mathbf{x}_t\|_2^2$ to reach the target accuracy $\epsilon > 0$, i.e., $\sin^2 \theta(\mathbf{x}_T, \mathbf{V}_p) < \epsilon$. The goal of our job here is to upper bound $T$ with the gap information and show that the bound is tight, namely

**Theorem 1** *The Riemannian gradient descent for Problem (1) with step-size* $\eta = \Theta(\frac{1}{\lambda_1 - \lambda_n}) \le \frac{1}{\lambda_1 - \lambda_n}$ *converges in* $T = \Theta(\frac{1}{\Delta} \log \frac{n}{\epsilon})$ *iterations, i.e.,* $\sin^2 \theta(\mathbf{x}_T, \mathbf{V}_p) < \epsilon$, *w.h.p. (with high probability).*

**Proof** First, it is trivial and $T = 0$ for the case that $\lambda_1 = \lambda_n$ since $f(\mathbf{x})$ is a constant. Thus, it is always assumed that $\lambda_1 > \lambda_n$ in what follows. Let $h_t(x) = 1 + \eta(x - \mathbf{x}_t^\top \mathbf{A}\mathbf{x}_t)$ and write $h_t(\mathbf{A}) = \sum_{i=1}^n h_t(\lambda_i)\mathbf{v}_i\mathbf{v}_i^\top = \mathbf{V}_n h_t(\mathbf{\Sigma_n})\mathbf{V}_n^\top$, where $h_t(\mathbf{\Sigma_n}) = \mathrm{diag}(h_t(\lambda_1), \cdots, h_t(\lambda_n))$. We can write that

$$\mathbf{x}_T = \frac{h_{T-1}(\mathbf{A})\mathbf{x}_{T-1}}{\|h_{T-1}(\mathbf{A})\mathbf{x}_{T-1}\|_2} = \cdots = \frac{\prod_{t=T-1}^0 h_t(\mathbf{A})\mathbf{x}_0}{\|\prod_{t=T-1}^0 h_t(\mathbf{A})\mathbf{x}_0\|_2} = \frac{\prod_{t=0}^{T-1} h_t(\mathbf{A})\mathbf{x}_0}{\|\prod_{t=0}^{T-1} h_t(\mathbf{A})\mathbf{x}_0\|_2}, \qquad (4)$$

where the second last equality is equivalently renormalized. Noting that the initial point $\mathbf{x}_0$ can be expressed as $\mathbf{x}_0 = \sum_{i=1}^n (\mathbf{v}_i^\top \mathbf{x}_0)\mathbf{v}_i$, the last numerator above can be written as

$$\prod_{t=0}^{T-1} h_t(\mathbf{A})\mathbf{x}_0 = \sum_{i=1}^n (\mathbf{v}_i^\top \mathbf{x}_0) \prod_{t=0}^{T-1} h_t(\mathbf{A})\mathbf{v}_i = \sum_{i=1}^n (\mathbf{v}_i^\top \mathbf{x}_0) \prod_{t=0}^{T-1} h_t(\lambda_i)\mathbf{v}_i.$$

Accordingly, it holds that

$$\|\mathbf{V}_p^\top \prod_{t=0}^{T-1} h_t(\mathbf{A})\mathbf{x}_0\|_2^2 = \sum_{i=1}^p (\mathbf{v}_i^\top \mathbf{x}_0)^2 \prod_{t=0}^{T-1} h_t^2(\lambda_i),$$

$$\|\prod_{t=0}^{T-1} h_t(\mathbf{A})\mathbf{x}_0\|_2^2 = \sum_{i=1}^n (\mathbf{v}_i^\top \mathbf{x}_0)^2 \prod_{t=0}^{T-1} h_t^2(\lambda_i).$$

By the fact that $\mathbf{x}^\top \mathbf{A}\mathbf{x} \in [\lambda_n, \lambda_1]$ for $\mathbf{x} \in \mathbb{S}^{n-1}$ (Corollary 4.7 in [15]), we have that $h_t(\lambda_i) \geq 1 + \eta(\lambda_n - \lambda_1)$ for all $i$ and $t$. If $1 + \eta(\lambda_n - \lambda_1) \geq 0$, i.e., $\eta \leq \frac{1}{\lambda_1 - \lambda_n}$, it then holds that $h_t(\lambda_i) \geq 0$ for all $i$ and $t$. When $\eta \leq \frac{1}{\lambda_1 - \lambda_n}$, we can get that

$$
\begin{aligned}
\sin^2 \theta(\mathbf{x}_T, \mathbf{V}_p) &= 1 - \|\mathbf{V}_p^\top \mathbf{x}_T\|_2^2 \\
&= 1 - \frac{\sum_{i=1}^p (\mathbf{v}_i^\top \mathbf{x}_0)^2 \prod_{t=0}^{T-1} h_t^2(\lambda_i)}{\sum_{i=1}^n (\mathbf{v}_i^\top \mathbf{x}_0)^2 \prod_{t=0}^{T-1} h_t^2(\lambda_i)} = \frac{\sum_{i=p+1}^n (\mathbf{v}_i^\top \mathbf{x}_0)^2 \prod_{t=0}^{T-1} h_t^2(\lambda_i)}{\sum_{i=1}^n (\mathbf{v}_i^\top \mathbf{x}_0)^2 \prod_{t=0}^{T-1} h_t^2(\lambda_i)} \qquad (5) \\
&\leq \frac{\prod_{t=0}^{T-1} h_t^2(\lambda_{p+1}) \sum_{i=p+1}^n (\mathbf{v}_i^\top \mathbf{x}_0)^2}{\prod_{t=0}^{T-1} h_t^2(\lambda_1) \sum_{i=1}^p (\mathbf{v}_i^\top \mathbf{x}_0)^2} = \prod_{t=0}^{T-1} (\frac{h_t(\lambda_{p+1})}{h_t(\lambda_1)})^2 \tan^2 \theta(\mathbf{x}_0, \mathbf{V}_p) \\
&\leq (\frac{1 + \eta(\lambda_{p+1} - \lambda_n)}{1 + \eta(\lambda_1 - \lambda_n)})^{2T} \tan^2 \theta(\mathbf{x}_0, \mathbf{V}_p) \\
&\leq \exp\{-\frac{2\eta\Delta_p T}{1 + \eta(\lambda_1 - \lambda_n)}\} \tan^2 \theta(\mathbf{x}_0, \mathbf{V}_p) \triangleq \epsilon,
\end{aligned}
$$

where the second last inequality have used the fact that $\frac{1+\eta(\lambda_{p+1}-x)}{1+\eta(\lambda_1-x)}$ is nonnegative and monotonically decreasing for $x \in [\lambda_n, \lambda_1]$. Solving the last equality about $\epsilon$ for $T$ yields that $T = \frac{1+\eta(\lambda_1-\lambda_n)}{2\eta\Delta_p} \log \frac{\tan^2 \theta(\mathbf{x}_0, \mathbf{V}_p)}{\epsilon}$. Note that $\mathbf{V}_{-p}^\top \widehat{\mathbf{x}}_0$ and $(\mathbf{V}_p^\top \widehat{\mathbf{x}}_0)^\dagger$ are entry-wise standard Gaussian since $\mathbf{V}_{-p}$ and $\mathbf{V}_p$ are orthonormal. By standard Gaussian matrix concentration theory [12], w.h.p. $\|\mathbf{V}_{-p}^\top \widehat{\mathbf{x}}_0\|_2^2 \leq c_1 n$ and $\|(\mathbf{V}_p^\top \widehat{\mathbf{x}}_0)^\dagger\|_2^2 \leq c_2 p$ for some fixed constants $c_1, c_2$. We then have w.h.p. that $\tan^2 \theta(\mathbf{x}_0, \mathbf{V}_p) \leq cnp$ for some fixed constant $c$ and $\log \tan^2 \theta(\mathbf{x}_0, \mathbf{V}_p) = O(\log(n))$. Hence, $T = O(\frac{1}{\Delta} \log \frac{n}{\epsilon})$ for $\eta = \Theta(\frac{1}{\lambda_1 - \lambda_n})$.

To justify the tightness of this upper bound, we now show that the Riemannian gradient descent for Problem (1) requires $\Omega(\frac{1}{\Delta} \log \frac{n}{\epsilon})$ iterations to converge. To this end, we resume the analysis from Eq. (5) as follows:

$$
\begin{aligned}
\sin^2 \theta(\mathbf{x}_T, \mathbf{V}_p) &= 1 - \|\mathbf{V}_p^\top \mathbf{x}_T\|_2^2 = \frac{\sum_{i=p+1}^n (\mathbf{v}_i^\top \mathbf{x}_0)^2 \prod_{t=0}^{T-1} h_t^2(\lambda_i)}{\sum_{i=1}^n (\mathbf{v}_i^\top \mathbf{x}_0)^2 \prod_{t=0}^{T-1} h_t^2(\lambda_i)} \\
&\geq \frac{(\mathbf{v}_{p+1}^\top \mathbf{x}_0)^2 \prod_{t=0}^{T-1} h_t^2(\lambda_{p+1})}{\sum_{i=1}^{p+1} (\mathbf{v}_i^\top \mathbf{x}_0)^2 \prod_{t=0}^{T-1} h_t^2(\lambda_i)} = \frac{1}{1 + \frac{\sum_{i=1}^p (\mathbf{v}_i^\top \mathbf{x}_0)^2}{(\mathbf{v}_{p+1}^\top \mathbf{x}_0)^2} \cdot \prod_{t=0}^{T-1} (\frac{h_t(\lambda_1)}{h_t(\lambda_{p+1})})^2} \\
&\geq \frac{1}{1 + \frac{\sum_{i=1}^p (\mathbf{v}_i^\top \mathbf{x}_0)^2}{(\mathbf{v}_{p+1}^\top \mathbf{x}_0)^2} \cdot \prod_{t=0}^{T-1} (\frac{1}{1-\eta\Delta_p})^2} = \frac{(1 - \eta\Delta_p)^{2T} \tan^2 \hat{\theta}}{1 + (1 - \eta\Delta_p)^{2T} \tan^2 \hat{\theta}} \\
&\geq \frac{\exp\{-2\eta\Delta_p T/(1-\eta\Delta_p)\} \tan^2 \hat{\theta}}{1 + \exp\{-2\eta\Delta_p T/(1-\eta\Delta_p)\} \tan^2 \hat{\theta}},
\end{aligned}
$$

where $\tan^2 \hat{\theta} = \frac{(\mathbf{v}_{p+1}^\top \mathbf{x}_0)^2}{\sum_{i=1}^p (\mathbf{v}_i^\top \mathbf{x}_0)^2}$ and we have used for the last inequality that $(1 - \eta \Delta_p)^{2T} =$ $\exp\{2T \log(1 - \eta \Delta_p)\} \geq \exp\{-\frac{2\eta \Delta_p T}{1 - \eta \Delta_p}\}$ by $\log(1 + x) \geq \frac{x}{1+x}$ for $x > -1$. Setting $\exp\{-\frac{2\eta \Delta_p T}{1 - \eta \Delta_p}\} \tan^2 \hat{\theta} = 2\epsilon$ yields that $T = \frac{1 - \eta \Delta_p}{2\eta \Delta_p} \log \frac{\tan^2 \hat{\theta}}{2\epsilon}$ and $\sin^2 \theta(\mathbf{x}_T, \mathbf{V}_p) \geq \frac{2\epsilon}{1+2\epsilon} \geq \epsilon$, assuming $\epsilon < \frac{1}{2}$. Hence, for $\eta = \Theta(\frac{1}{\lambda_1 - \lambda_n})$, we need $T = \Omega(\frac{1}{\Delta} \log \frac{n}{\epsilon})$ iterations to get $\sin^2 \theta(\mathbf{x}_T, \mathbf{V}_p) < \epsilon$, which completes the proof. $\qquad\square$

**Remark 1** In the gap-dependent analysis above, from the optimization point of view, we care more about the distance of the iterate $\mathbf{x}_t$ to the optimal solution set $\mathcal{V}$, i.e., $\sin^2 \theta(\mathbf{x}_t, \mathbf{V}_p) = \frac{1}{2} \min_{\mathbf{v} \in \mathcal{V}} \|\mathbf{x}_t - \mathbf{v}\|_2^2$, with emphasis on the eigenvector computation. In fact, it is also common to consider the objective sub-optimality gap, i.e., $(f(\mathbf{x}_t) - \min_{\mathbf{x} \in \mathbb{S}^{n-1}} f(\mathbf{x})) = \frac{1}{2}(\lambda_1 - \mathbf{x}_t^\top \mathbf{A} \mathbf{x}_t)$, with emphasis on the eigenvalue computation. From the proof of Lemma 4.6 in Xu and Gao 2018 [20], we know that $\lambda_1 - \mathbf{x}_T^\top \mathbf{A} \mathbf{x}_T \leq (\lambda_1 - \lambda_n) \sin^2 \theta(\mathbf{x}_T, \mathbf{V}_p)$. By Theorem 1, with $\epsilon$-rescaling, we then have that $\lambda_1 - \mathbf{x}_T^\top \mathbf{A} \mathbf{x}_T < \epsilon$ for $T = O(\frac{1}{\Delta} \log \frac{n}{\epsilon})$.

# 5 Gap-Independent Analysis

The issue of the gap-dependent rate in Theorem 1 is that when the gap $\Delta$ is significantly smaller than the target accuracy $\epsilon$, the convergence in terms of $\sin^2 \theta(\mathbf{x}_T, \mathbf{V}_p)$ is too slow, e.g., $T = \Theta(\frac{1}{\epsilon^2} \log \frac{n}{\epsilon})$ for $\Delta = \Theta(\epsilon^2)$. In this case, it is often acceptable to swap the true leading eigenvector for any $\epsilon$-accurate solution $\mathbf{x}_T \in \mathbb{S}^{n-1}$ in terms of the objective sub-optimality gap, i.e., $\lambda_1 - \mathbf{x}_T^\top \mathbf{A} \mathbf{x}_T < \epsilon$. In this section, by extending the analysis of Musco et al. 2015 [10] for constant matrices (i.e., $h_t(x) = x$), we show without the need of the eigenvalue gap information that even if the gap $\Delta$ is significantly smaller than the target accuracy, we only need $O(\frac{1}{\epsilon} \log \frac{n}{\epsilon})$ (far less than $O(\frac{1}{\Delta} \log \frac{n}{\epsilon})$) for, e.g., $\Delta = \Theta(\epsilon^{3/2})$, see Remark 1) iterations to achieve that $\lambda_1 - \mathbf{x}_T^\top \mathbf{A} \mathbf{x}_T < \epsilon$, as stated in the following theorem on the worst-case performance.

**Theorem 2** *The Riemannian gradient descent for Problem (1) with step-size $\eta = O(1) \leq \frac{1}{\lambda_1 - \lambda_n}$ converges in $T = O(\frac{1}{\epsilon} \log \frac{n}{\epsilon})$ iterations, i.e., $\lambda_1 - \mathbf{x}_T^\top \mathbf{A} \mathbf{x}_T < \epsilon$.*

**Proof Sketch** Due to space limit, we only give proof sketch here. The complete proof and all the missing proofs can be found in the supplementary material. We assume again that $\lambda_1 > \lambda_n$, and $\eta \leq \frac{1}{\lambda_1 - \lambda_n}$ such that $h_t(\lambda_i) = 1 + \eta(\lambda_i - \mathbf{x}_t^\top \mathbf{A} \mathbf{x}_t) \geq 0$ for all $i$ and $t$. It suffices to show that $\lambda_1 - \mathbf{x}_T^\top \mathbf{A} \mathbf{x}_T < \frac{2}{\eta} \epsilon$ always holds no matter whether $\lambda_1$ is significantly larger than $\lambda_2$ in the sense that $h_0(\lambda_1) \geq (1 + \delta/2) h_0(\lambda_2)$ for $0 < \delta \leq 2$. We take $T = \lceil \frac{2}{\delta} \log \frac{n(1 + \tan^2 \theta_0)}{\epsilon} \rceil + 1$, where $\theta_0 = \theta(\mathbf{x}_0, \mathbf{v}_1)$.

$\boxed{\text{Case 1}}$ that $h_0(\lambda_1) \geq (1 + \delta/2) h_0(\lambda_2)$. Consider the polynomial

$$p_T(x) = \sqrt{(1 + \delta/2) h_0(\lambda_2)} \prod_{t=0}^{T-1} \frac{h_t(x)}{(1 + \delta/2) h_t(\lambda_2)} \geq 0, \quad x \in [\lambda_n, \lambda_1]$$

and matrix form $p_T(\mathbf{A}) = \sum_{i=1}^n p_T(\lambda_i) \mathbf{v}_i \mathbf{v}_i^\top = \mathbf{V}_n p_T(\mathbf{\Sigma}_n) \mathbf{V}_n^\top$, where $p_T(\mathbf{\Sigma}_n) = \mathrm{diag}(p_T(\lambda_1), \cdots, p_T(\lambda_n))$. We have

**Fact 1.** For $x \in [\lambda_n, \lambda_2]$, $h_0(\lambda_1) \geq (1 + \delta/2) h_0(x)$ implies $h_t(\lambda_1) \geq (1 + \delta/2) h_t(x)$ for all $t$, by

**Lemma 3** *If $\eta \leq \frac{2}{\lambda_1 - \lambda_n}$ then $\mathbf{x}_{t+1}^\top \mathbf{A} \mathbf{x}_{t+1} \geq \mathbf{x}_t^\top \mathbf{A} \mathbf{x}_t$.*

Hence, $p_T(\lambda_1) \geq \sqrt{h_0(\lambda_1)}$. Also, $p_T(\lambda_i) \leq \sqrt{h_0(\lambda_2)}(1 + \delta/2)^{-T + \frac{1}{2}}, i = 2, \cdots, n$. Eq. (4) can be rewritten as $\mathbf{x}_T = \frac{p_T(\mathbf{A}) \mathbf{x}_0}{\|p_T(\mathbf{A}) \mathbf{x}_0\|_2}$, where the rank-1 approximation inequality for $p_T(\mathbf{A})$ holds:

$$
\begin{aligned}
\|p_T(\mathbf{A}) - \mathbf{x}_T \mathbf{x}_T^\top p_T(\mathbf{A})\|_F^2 &\leq (1 + \tan^2 \theta_0) \|p_T(\mathbf{A}) - p_T(\lambda_1) \mathbf{v}_1 \mathbf{v}_1^\top\|_F^2 \\
&= (1 + \tan^2 \theta_0) \sum_{i=2}^n p_T^2(\lambda_i) \\
&\leq (1 + \tan^2 \theta_0)(n-1) h_0(\lambda_2)(1 + \delta/2)^{-2T+1} < h_0(\lambda_2) \epsilon.
\end{aligned}
$$

At the same time,

$$\|p_T(\mathbf{A}) - \mathbf{x}_T\mathbf{x}_T^\top p_T(\mathbf{A})\|_F^2 = \|p_T(\mathbf{A})\|_F^2 - \|\mathbf{x}_T\mathbf{x}_T^\top p_T(\mathbf{A})\|_F^2 = \sum_{i=1}^n (1 - (\mathbf{x}_T^\top \mathbf{v}_i)^2)p_T^2(\lambda_i)$$
$$\geq (1 - (\mathbf{x}_T^\top \mathbf{v}_1)^2)p_T^2(\lambda_1) \geq (1 - (\mathbf{x}_T^\top \mathbf{v}_1)^2)h_0(\lambda_1).$$

Thus, $\lambda_1 - \mathbf{x}_T^\top \mathbf{A}\mathbf{x}_T < \frac{2}{\eta}\epsilon$ holds by noting that

$$h_0(\lambda_1) - \mathbf{x}_T^\top h_0(\mathbf{A})\mathbf{x}_T = h_0(\lambda_1) - \sum_{i=1}^n (\mathbf{x}_T^\top \mathbf{v}_i)^2 h_0(\lambda_i) \leq (1 - (\mathbf{x}_T^\top \mathbf{v}_1)^2)h_0(\lambda_1) < \epsilon h_0(\lambda_2).$$

$\boxed{\text{Case 2}}$ that $h_0(\lambda_1) < (1 + \delta/2)h_0(\lambda_2)$. Consider the polynomial $q_T(x) = \sqrt{h_0(\lambda_1)}\prod_{t=0}^{T-1}\frac{h_t(x)}{h_t(\lambda_1)}$. and write $\mathbf{x}_T = \frac{q_T(\mathbf{A})\mathbf{x}_0}{\|q_T(\mathbf{A})\mathbf{x}_0\|_2}$. Define the index set $\alpha = \{i : \frac{1}{1+\delta/2}h_0(\lambda_1) \leq h_0(\lambda_i) < h_0(\lambda_1)\}$ and rewrite

$$\mathbf{x}_T = \frac{\sum_{i\in\alpha} q_T(\lambda_i)\mathbf{v}_i\mathbf{v}_i^\top \mathbf{x}_0}{\|q_T(\mathbf{A})\mathbf{x}_0\|_2} + \frac{\sum_{i\notin\alpha} q_T(\lambda_i)\mathbf{v}_i\mathbf{v}_i^\top \mathbf{x}_0}{\|q_T(\mathbf{A})\mathbf{x}_0\|_2} \triangleq \tilde{\mathbf{x}}_T^{(\alpha)} + \tilde{\mathbf{x}}_T^{(-\alpha)}.$$

We then have $\|h_0^{\frac{1}{2}}(\mathbf{A})\mathbf{x}_T^{(\alpha)}\|_2^2 \geq \frac{1}{1+\delta/2}h_0(\lambda_1) \geq (1 - \epsilon)h_0(\lambda_1)$ for $\delta = 2\epsilon$, where the first inequality is by the definition of the index set. For $\mathbf{x}_T^{(-\alpha)} \triangleq \frac{\tilde{\mathbf{x}}_T^{(-\alpha)}}{\|\tilde{\mathbf{x}}_T^{(-\alpha)}\|_2} = \frac{q_T(\mathbf{A}_{-\alpha})\mathbf{x}_0}{\|q_T(\mathbf{A}_{-\alpha})\mathbf{x}_0\|_2}$ and $\mathbf{A}_{-\alpha} \triangleq \sum_{i\notin\alpha}\lambda_i\mathbf{v}_i\mathbf{v}_i^\top$, a similar argument to Case 1 can be applied to get that $h_0(\lambda_1) - (\mathbf{x}_T^{(-\alpha)})^\top h_0(\mathbf{A}_{-\alpha})\mathbf{x}_T^{(-\alpha)} < h_0(\lambda_1)\epsilon$. Then $\|h_0^{\frac{1}{2}}(\mathbf{A})\mathbf{x}_T^{(-\alpha)}\|_2^2 = (\mathbf{x}_T^{(-\alpha)})^\top h_0(\mathbf{A}_{-\alpha})\mathbf{x}_T^{(-\alpha)} > (1 - \epsilon)h_0(\lambda_1)$. Thus, $\lambda_1 - \mathbf{x}_T^\top \mathbf{A}\mathbf{x}_T < \frac{2}{\eta}\epsilon$ holds by noting that

$$\mathbf{x}_T^\top h_0(\mathbf{A})\mathbf{x}_T = \|h_0^{\frac{1}{2}}(\mathbf{A})\tilde{\mathbf{x}}_T^{(\alpha)}\|_2^2 + \|h_0^{\frac{1}{2}}(\mathbf{A})\tilde{\mathbf{x}}_T^{(-\alpha)}\|_2^2$$
$$= \|\tilde{\mathbf{x}}_T^{(\alpha)}\|_2^2\|h_0^{\frac{1}{2}}(\mathbf{A})\mathbf{x}_T^{(\alpha)}\|_2^2 + \|\tilde{\mathbf{x}}_T^{(-\alpha)}\|_2^2\|h_0^{\frac{1}{2}}(\mathbf{A})\mathbf{x}_T^{(-\alpha)}\|_2^2$$
$$> (\|\tilde{\mathbf{x}}_T^{(\alpha)}\|_2^2 + \|\tilde{\mathbf{x}}_T^{(-\alpha)}\|_2^2)(1 - \epsilon)h_0(\lambda_1) = (1 - \epsilon)h_0(\lambda_1).$$

$\square$

**Remark 2** When the gap is significantly larger than the target accuracy $\epsilon$, e.g., $\Delta = O(1)$, and $\sin^2\theta(\mathbf{x}_T, \mathbf{V}_p) \geq \frac{\epsilon}{(\lambda_1-\lambda_n)\Delta}$ for $T = \Theta(\frac{1}{\Delta}\log\frac{n(\lambda_1-\lambda_n)\Delta}{\epsilon})$ (see the proof of Theorem 1 on the tightness), we have that $\lambda_1 - \mathbf{x}_T^\top \mathbf{A}\mathbf{x}_T \geq (\lambda_1 - \lambda_n)\Delta\sin^2\theta(\mathbf{x}_T, \mathbf{V}_p) \geq \epsilon$ by Lemma 2 in Xu et al. [19]. Thus, we need $T = \Omega(\frac{1}{\Delta}\log\frac{n}{\epsilon})$ iterations to get $\lambda_1 - \mathbf{x}_T^\top \mathbf{A}\mathbf{x}_T < \epsilon$. Otherwise, it needs $O(\frac{1}{\epsilon}\log\frac{n}{\epsilon})$ iterations to make it. Thus, a comprehensively tight rate can be stated as $O(\frac{1}{\max\{\Delta,\epsilon\}}\log\frac{n}{\epsilon})$.

**Remark 3** The analysis of Case 1 in the proof of Theorem 2 can give us a gap-dependent rate:

$$h_0(\lambda_1) = 1 + \eta(\lambda_1 - \mathbf{x}_0^\top \mathbf{A}\mathbf{x}_0) = (1 + \frac{\eta\Delta_1}{1+\eta(\lambda_2 - \mathbf{x}_0^\top \mathbf{A}\mathbf{x}_0)})(1 + \eta(\lambda_2 - \mathbf{x}_0^\top \mathbf{A}\mathbf{x}_0))$$
$$\geq (1 + \frac{\eta\Delta_1}{1+\eta(\lambda_2 - \lambda_n)})h_0(\lambda_2) \geq (1 + \frac{\eta\Delta_1}{2})h_0(\lambda_2).$$

Taking $\delta = \eta\Delta_1$ in $T = \lceil\frac{2}{\delta}\log\frac{n(1+\tan^2\theta_0)}{\epsilon}\rceil + 1$ gives us $T = O(\frac{1}{\eta\Delta_1}\log\frac{n}{\epsilon})$. It is straightforward to extend to the setting $\lambda_1 = \cdots = \lambda_p > \lambda_{p+1}$ of the previous section to get $T = O(\frac{1}{\Delta}\log\frac{n}{\epsilon})$ with $\Delta_p$ and $\eta = \Theta(\frac{1}{\lambda_1-\lambda_n})$. But it is worth mentioning this is about the convergence in terms of $\lambda_1 - \mathbf{x}_T^\top \mathbf{A}\mathbf{x}_T < \epsilon$, rather than $\sin^2\theta(\mathbf{x}_T, \mathbf{V}_p) < \epsilon$.

**Remark 4** Compared to $h_t(\mathbf{A}) \equiv \mathbf{A}$ which leads to simple monomials in [7, 10], $h_t(\mathbf{A}) = \mathbf{I} + \eta(\mathbf{A} - \mathbf{x}_t^\top \mathbf{A}\mathbf{x}_t \cdot \mathbf{I})$ varies with iterations and requires complicated polynomials in our analysis. In addition, from Remark 1-2, it holds for $\|\mathbf{x}\|_2 = 1$ that $\frac{1}{2}\Delta_p\sin^2\theta(\mathbf{x}, \mathbf{V}_p) \leq f(\mathbf{x}) - f(\mathbf{v}_1) \leq \frac{1}{2}(\lambda_1 - \lambda_n)\sin^2\theta(\mathbf{x}, \mathbf{V}_p)$. Note that either these inequalities or Wedin's theorem can't be used to translate results between Theorem 1 and Theorem 2, because the translated results by these inequalities may possibly be worse and Wedin's theorem (which upper bounds perturbation of eigenspace in

subspace distance by those of both the given matrix and corresponding gap) is inapplicable here. For example, by the second inequality and Theorem 1, we can get $\lambda_1 - \mathbf{x}_T \mathbf{A} \mathbf{x}_T < \epsilon$ for $T = O(\frac{1}{\Delta} \log \frac{n}{\epsilon})$. But when $\Delta$ is much less than $\epsilon$, $T$ will clearly be worse than the results in Theorem 2. On the other hand, by the first inequality and Theorem 2, we can get $\sin^2 \theta(\mathbf{x}_T, \mathbf{V}_p) < \epsilon$ for $T = O(\frac{1}{\epsilon \Delta_p} \log \frac{n}{\epsilon \Delta_p})$, which is worse than the results in Theorem 1.

## 6 Projection Versus Search

To be specific, in this section, we refer to the power iteration (PI) and Riemannian gradient descent (RGD) as the projection and search methods for Problem (1), respectively, and the projected gradient descent (PGD) belongs to both classes.

The above analyses for the search method apply to the PGD by simply setting $h_t(x) = 1 + \eta x$. In this case, $\eta$ represents the step-size and needs satisfy $0 < \eta \le \frac{1}{\rho}$ to make sure $h_t(\lambda_i) \ge 0$ for all $i$, where $\rho = \max_i |\lambda_i|$ is $\mathbf{A}$'s spectral radius. Assuming that $\Delta_1 > 0$, it will converge to $\mathbf{v}_1$. On the other hand, the analyses [7, 10] of the PI apply to the PGD as well. Then $\eta$ can be any nonzero real value and the PGD converges to $\mathbf{v}_{\sigma(1)}$ assuming that the largest two values of $\{|h_t(\lambda_j)| : j = 1, \cdots, n\}$ are distinct, where $\sigma$ maps $i$ to $j$ such that $|h_t(\lambda_j)|$ is the $i$-th largest value of $\{|h_t(\lambda_j)| : j = 1, \cdots, n\}$. In addition, the PI can be seen as the PGD with constant step-size $\eta = \pm \infty$, and can also be interpreted as the RGD with varying step-sizes $\eta_t = \frac{1}{\mathbf{x}_t^\top \mathbf{A} \mathbf{x}_t}$ when $\mathbf{A}$ is positive definite.

Table 2: Comparison of the PI, PGD, and RGD.

| | $\mathbf{x}_{t+1}$ | $\eta$ | $\Delta$ | $T$ | $\mathbf{x}_T$ | $\mathbf{x}_T^\top \mathbf{A} \mathbf{x}_T$ |
|---|---|---|---|---|---|---|
| PI | $\frac{(\mathbf{I}+\eta\mathbf{A})\mathbf{x}_t}{\|(\mathbf{I}+\eta\mathbf{A})\mathbf{x}_t\|_2}$ | $\{\pm\infty\}$ | $1-\left|\frac{\lambda_{\sigma(2)}}{\lambda_{\sigma(1)}}\right|$ | $O(\frac{1}{\max\{\Delta,\epsilon\}} \log \frac{n}{\epsilon})$ | $\mathbf{v}_{\sigma(1)}$ | $\lambda_{\sigma(1)}$ |
| PGD | $\frac{\mathbf{x}_t+\eta\nabla f(\mathbf{x}_t)}{\|\mathbf{x}_t+\eta\nabla f(\mathbf{x}_t)\|_2}$ | $(0, \frac{1}{\rho}]$ | $\frac{\Delta_1}{\rho}$ | $O(\frac{1}{\max\{\Delta,\epsilon\}} \log \frac{n}{\epsilon})$ | $\mathbf{v}_1$ | $\lambda_1$ |
| RGD | $\frac{\mathbf{x}_t+\eta\tilde{\nabla} f(\mathbf{x}_t)}{\|\mathbf{x}_t+\eta\tilde{\nabla} f(\mathbf{x}_t)\|_2}$ | $(0, \frac{1}{\lambda_1 - \lambda_n}]$ | $\frac{\Delta_1}{\lambda_1 - \lambda_n}$ | $O(\frac{1}{\max\{\Delta,\epsilon\}} \log \frac{n}{\epsilon})$ | $\mathbf{v}_1$ | $\lambda_1$ |

Table 3: Contraction factors of the PI, PGD, and RGD on positive definite matrices.

| | PI | PGD | RGD |
|---|---|---|---|
| $h_t(x)$ | $x$ | $1 + \eta x$ | $1 + \eta(x - \mathbf{x}_t^\top \mathbf{A} \mathbf{x}_t)$ |
| $\frac{h_t(\lambda_2)}{h_t(\lambda_1)}$ | $1 - \frac{\Delta_1}{\lambda_1}$ | $1 - \frac{\Delta_1}{\frac{1}{\eta}+\lambda_1}$ | $1 - \frac{\Delta_1}{\frac{1}{\eta}+\lambda_1-\mathbf{x}_t^\top \mathbf{A} \mathbf{x}_t}$ |

Table 2 summarizes the comparisons between these methods for Problem (1), where $\epsilon$ represents the target accuracy for $\lambda_1 - \mathbf{x}_T^\top \mathbf{A} \mathbf{x}_T$, and $(\mathbf{x}_T^\top \mathbf{A} \mathbf{x}_T, \mathbf{x}_T)$ describes which eigenpair it approximates. Table 3 shows that the PGD even with varying step-sizes $0 < \eta_t \le \frac{1}{\rho}$ always gets a larger contraction factor $\frac{h_t(\lambda_2)}{h_t(\lambda_1)}$ (see the equation right below Eq. (5)) than the PI, but the RGD can achieve a smaller one when step-sizes satisfy that $\eta_t > \frac{1}{\mathbf{x}_t^\top \mathbf{A} \mathbf{x}_t}$. In practice, it may be worth a try to set $\eta_t = \frac{\tau}{\mathbf{x}_t^\top \mathbf{A} \mathbf{x}_t}$ with $\tau > 1$. However, the Riemannian gradient descent update requires more computations than the power iteration in each step, possibly leading to an even worse wall-clock running time. This can be easily seen from the equivalence between the two methods with $\eta_t = \frac{1}{\mathbf{x}_t^\top \mathbf{A} \mathbf{x}_t}$.

In the next section, we will investigate the effect of this step-size scheme $\eta_t = \frac{\tau}{\mathbf{x}_t^\top \mathbf{A} \mathbf{x}_t}$ ($\tau > 1$) on both projected and Riemannian gradient descent for eigenvector computation as well as the stochastic variance reduced algorithms for PCA (VR-PCA) based on Oja's algorithm or Krasulina's method.

# 7 Experiments

The purpose of the experimental study for corroborating our findings in above sections is twofold. One is to verify the rate $O((1/\max\{\Delta, \epsilon\}) \log(n/\epsilon))$ of the Riemannian gradient descent on real symmetric matrices in the form of $\mathbf{A} \in \mathbb{R}^{n \times n}$. The other is to experiment on the step-size scheme for both projected and Riemannian gradient descent methods on PCA data in the form of $\mathbf{C} \in \mathbb{R}^{d \times n}$ such that $\mathbf{A} = \frac{1}{n} \mathbf{C} \mathbf{C}^\top$. Experiments were done on a laptop (dual-core 2.30GHZ CPU and 8GB RAM).

The constituent part $O((1/\Delta) \log(n/\epsilon))$ of the rate has been experimentally verified in Ding et al. [5]. Hence, only the worst-case rate $O((1/\epsilon) \log(n/\epsilon))$ needs verification, for which we generate synthetic data $\mathbf{A} = \mathbf{V} \mathbf{\Sigma} \mathbf{V}^\top$ as follows. Given $n = 1000, s = 100, \Delta = 0.01$ and entry-wise i.i.d. standard normal $\mathbf{U} \in \mathbb{R}^{n \times n}, \mathbf{a} \in \mathbb{R}^{(n-s-1) \times 1}$, we consider orthogonal $\mathbf{V} = \mathbf{U}(\mathbf{U}^\top \mathbf{U})^{-1/2}$ and diagonal $\mathbf{\Sigma} = \mathrm{diag}(2\mathbf{I} - \Delta \mathrm{diag}(0, 1, \cdots, s), \frac{1}{n} \mathrm{diag}(|a_1|, \cdots, |a_{n-s-1}|))$. In this case, the convergence for each method considered in this work should behave differently in terms of different measures, i.e., $O((1/\epsilon) \log(n/\epsilon))$ for $(\lambda_1 - \mathbf{x}_T^\top \mathbf{A} \mathbf{x}_T)$ versus $O((1/\Delta) \log(n/\epsilon))$ for $\sin^2 \theta(\mathbf{x}_T, \mathbf{v}_1)$.

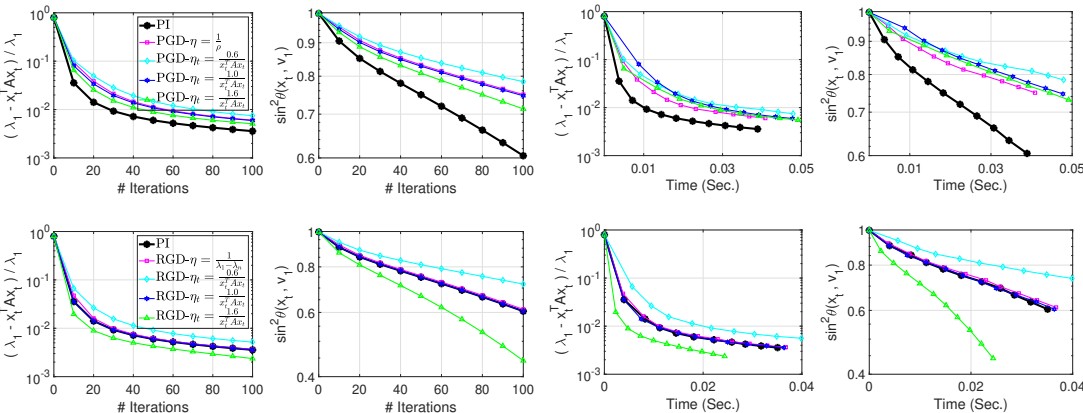

Figure 1: Test on synthetic data.

We implemented the PGD with $\eta = \frac{1}{\rho}, \eta_t = \frac{0.6}{\mathbf{x}_t^\top \mathbf{A} \mathbf{x}_t}, \frac{1}{\mathbf{x}_t^\top \mathbf{A} \mathbf{x}_t}, \frac{1.6}{\mathbf{x}_t^\top \mathbf{A} \mathbf{x}_t}$, and RGD with step-size schemes $\eta = \frac{1}{\lambda_1 - \lambda_n}, \eta_t = \frac{0.6}{\mathbf{x}_t^\top \mathbf{A} \mathbf{x}_t}, \frac{1}{\mathbf{x}_t^\top \mathbf{A} \mathbf{x}_t}, \frac{1.6}{\mathbf{x}_t^\top \mathbf{A} \mathbf{x}_t}$, in MATLAB. Each is compared to the PI. All the methods start from the same random initial point $\mathbf{x}_0$ and run for $T = 100$ iterations. Figure 1 reports their performance across iterations (or the average performance over time on 10 runs) in two measures, where the y-axis of the figures in the first and third columns is in log scale of $(\lambda_1 - \mathbf{x}_t^\top \mathbf{A} \mathbf{x}_t)/\lambda_1$. It is clear for each method that $(\lambda_1 - \mathbf{x}_t^\top \mathbf{A} \mathbf{x}_t)$ converges sub-linearly while $\sin^2 \theta(\mathbf{x}_t, \mathbf{v}_1)$ is decreasing even more slowly, matching their respective theoretical rates. Consider the case that $\Delta = 0.01$ and accuracy $\epsilon = 0.1$ here. First, from the figures on the left in Figure 1, it is clear that the (relative) target in objective converges sub-linearly (note that y-axis is log scale here), roughly matching $O((1/\epsilon) \log(n/\epsilon))$ iterations. From the figures on the right, the target $\sin^2 \theta(\mathbf{x}_t, \mathbf{v}_1)$ converges about one order of magnitude slower than the target in objective, roughly matching $O((1/\epsilon)^2 \log(n/\epsilon))$ iterations which is basically consistent with $O((1/\Delta) \log(n/\epsilon))$ for $\epsilon = 0.1$.

We also experiment on two real datasets[4] **Schenk** that was used in [19, 5] and **GHS_indef**. Schenk is a $10728 \times 10728$ real symmetric matrix with 85000 nonzero entries. The performance is depicted in Figure 2, where all the methods converge linearly due to a large gap $\Delta$ and their differences are sharper. Particularly, we add the plot of the average performance over time on 10 runs. Although the RGD with $\eta_t = \frac{1}{\mathbf{x}_t^\top \mathbf{A} \mathbf{x}_t}$ is theoretically the same as the PI (thus performing exactly the same over iterations in Figures 1-2), it needs more computational costs reflected by the wall-clock time in Figure 2. Nonetheless, Figures 1-2 showcase the great potential of the RGD with aggressive step-size schemes to outperform the PI, e.g., $\eta_t = \frac{1.6}{\mathbf{x}_t^\top \mathbf{A} \mathbf{x}_t}$ here, in practice. In the meanwhile, the performance of the PGD even with adaptive step-sizes is worse than that of the PI. We tested significantly larger

---

[4] https://sparse.tamu.edu/

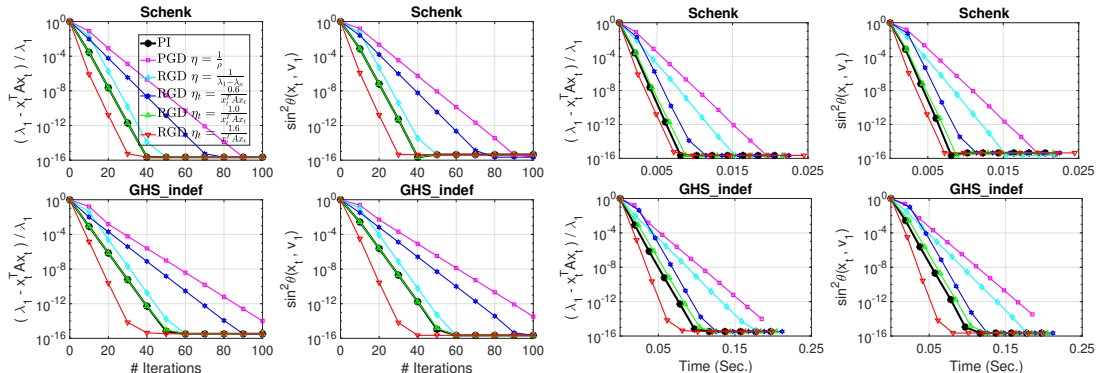

Figure 2: Deterministic Krasulina's method on Schenk and GHS_indef with adaptive step-sizes.

constant $\tau$ in $\frac{\tau}{\mathbf{x}_t^\top \mathbf{A} \mathbf{x}_t}$. PGD's performance is still worse than that of the PI. Similar results appear on GHS_indef which is a $76638 \times 76638$ real symmetric matrix with $859520$ nonzero entries.

| **Algorithm 1** VR-PCA (Oja) | **Algorithm 2** VR-PCA (Krasulina) |
|---|---|
| 1: **Input:** $\mathbf{C} = (\mathbf{c}_1, \cdots, \mathbf{c}_n) \in \mathbb{R}^{d \times n}$, $\tilde{\mathbf{x}}_0$, $\tau$, $b = |\alpha_t|$, $m = \lceil \frac{n}{b} \rceil$, $S$, where $[n] = \{1, 2, \cdots, n\}$ | 1: **Input:** $\mathbf{C} = (\mathbf{c}_1, \cdots, \mathbf{c}_n) \in \mathbb{R}^{d \times n}$, $\tilde{\mathbf{x}}_0$, $\tau$, $b = |\alpha_t|$, $m = \lceil \frac{n}{b} \rceil$, $S$ |
| 2: **Output:** $\mathbf{x}_S$ | 2: **Output:** $\mathbf{x}_S$ |
| 3: **for** $s = 0, 1, \cdots, S-1$ **do** | 3: **for** $s = 0, 1, \cdots, S-1$ **do** |
| 4: $\quad \mathbf{g}_s = \frac{1}{n}\mathbf{C}\mathbf{C}^\top \tilde{\mathbf{x}}_s$, $\mathbf{x}_0 = \tilde{\mathbf{x}}_s$, $\eta_s = \frac{n\tau}{\|\mathbf{C}^\top \tilde{\mathbf{x}}_s\|_2^2}$ | 4: $\quad \tilde{\mathbf{g}}_s = \frac{1}{n}(\mathbf{C}\mathbf{C}^\top \tilde{\mathbf{x}}_s - \|\mathbf{C}^\top \tilde{\mathbf{x}}_s\|_2^2 \tilde{\mathbf{x}}_s)$, $\mathbf{x}_0 = \tilde{\mathbf{x}}_s$, $\eta_s = \frac{n\tau}{\|\mathbf{C}^\top \tilde{\mathbf{x}}_s\|_2^2}$ |
| 5: $\quad$ **for** $t = 0, 1, \cdots, m-1$ **do** | 5: $\quad$ **for** $t = 0, 1, \cdots, m-1$ **do** |
| 6: $\quad\quad$ Pick $\alpha_t \subset [n]$ uniformly at random | 6: $\quad\quad$ Pick $\alpha_t \subset [n]$ uniformly at random |
| 7: $\quad\quad \mathbf{g}_t = \frac{1}{b}\sum_{i \in \alpha_t}\mathbf{c}_i\mathbf{c}_i^\top(\mathbf{x}_t - \tilde{\mathbf{x}}_s) + \mathbf{g}_s$ | 7: $\quad\quad \tilde{\mathbf{g}}_t = \frac{1}{b}\sum_{i \in \alpha_t}(\mathbf{c}_i\mathbf{c}_i^\top(\mathbf{x}_t - \tilde{\mathbf{x}}_s) - ((\mathbf{c}_i^\top \mathbf{x}_t)^2 \mathbf{x}_t - (\mathbf{c}_i^\top \tilde{\mathbf{x}}_s)^2 \tilde{\mathbf{x}}_s)) + \tilde{\mathbf{g}}_s$ |
| 8: $\quad\quad \mathbf{x}_{t+1} = \frac{\mathbf{x}_t + \eta_s \mathbf{g}_t}{\|\mathbf{x}_t + \eta_s \mathbf{g}_t\|_2}$ | 8: $\quad\quad \mathbf{x}_{t+1} = \frac{\mathbf{x}_t + \eta_s \tilde{\mathbf{g}}}{\|\mathbf{x}_t + \eta_s \tilde{\mathbf{g}}_t\|_2}$ |
| 9: $\quad$ **end for** | 9: $\quad$ **end for** |
| 10: $\quad \tilde{\mathbf{x}}_s = \mathbf{x}_m$ | 10: $\quad \tilde{\mathbf{x}}_s = \mathbf{x}_m$ |
| 11: **end for** | 11: **end for** |

Table 4: Statistics of PCA data

| DATA | DESCRIPTION | $d$ | $n$ |
|---|---|---|---|
| MMILL | MULTI-LABEL IMAGES | 221 | 30000 |
| JW11 | ACOUSTIC AND ARTICULATION | 385 | 30000 |
| MNIST | IMAGES OF HANDWRITTEN DIGITS | 784 | 70000 |

We now check the performance of two versions of the VR-PCA based on Oja's algorithm and Krasulina's method, corresponding to Algorithm 1 in the work of Shamir 2015 [14] (Algorithm 1 above) and variance reduced version (see the supplementary material) of Algorithm 1 in the work of Tang 2019 [16] (Algorithm 2 above), respectively, where the constant learning rate is replaced with the proposed adaptive learning rate scheme. The common PCA datasets are used and summarized in Table 4. We use $b = 100$. Note that we only update the learning rate at the epoch level and keep it unchanged within each epoch, similar to the case of the computation of the full gradient. Figure 3 reports the average performance of the VR-PCA with our adaptive learning rate on three real datasets, following the previous experimental setting on Schenk except for 5 runs used for average. It is easy to see that the proposed learning rate scheme (i.e., $\eta_s = \frac{\tau n}{\|\mathbf{C}\tilde{\mathbf{x}}_s\|_2^2}$) in the PCA setting consistently

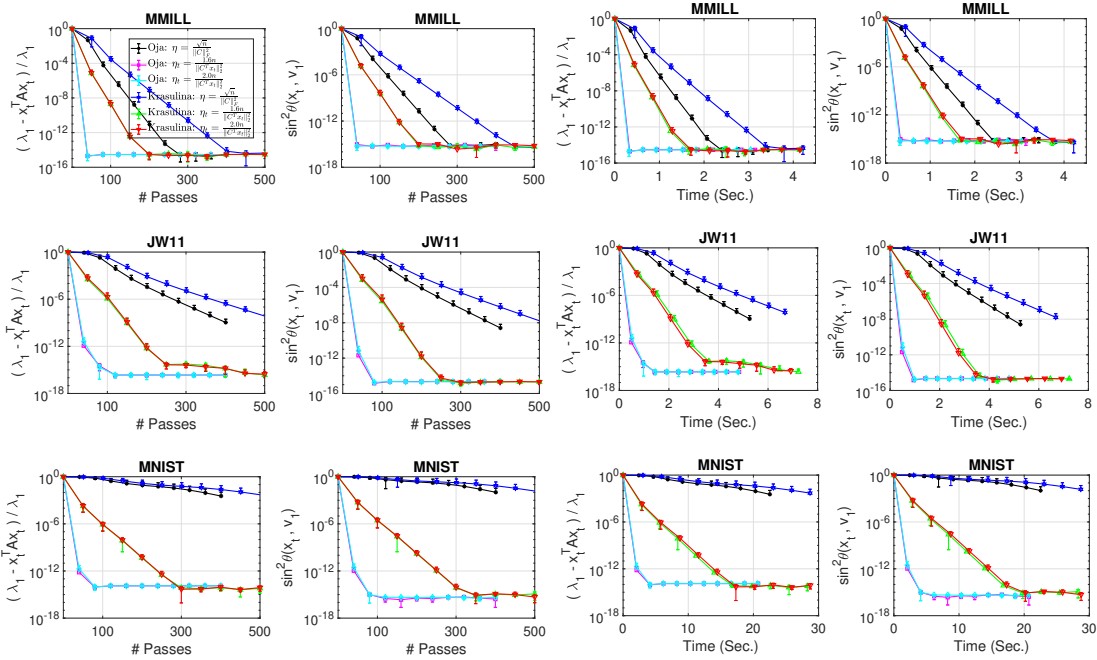

Figure 3: VR-PCA on MMILL, JW11, and MNIST with adaptive learning rates.

works far better than that proposed in Shamir 2015 [14] (i.e., $\eta = \frac{\sqrt{n}}{\|\mathbf{C}\|_F^2}$) across the datasets, for both versions of the VR-PCA. Interestingly, an opposite case (see Figure 2 for the deterministic setting) happens that the performance of the Oja-based VR-PCA performs significantly better than that based on Krasulina's method. This shows that the proposed learning rate scheme is very promising for the practitioners of the VR-PCA.

## 8    Conclusions

We presented two analyses of the Riemannian gradient descent for eigenvector computation. Specifically, the gap-dependent one is the first general analysis that achieves a tight rate $O(\frac{1}{\Delta} \log \frac{n}{\epsilon})$ in terms of the point distance to the optimal solutions. However, if we are only concerned with the objective distance to the optimal solutions, the rate of this type is not worst-case optimal. We succeeded in establishing a worst-case rate $O(\frac{1}{\epsilon} \log \frac{n}{\epsilon})$ independent of any gap, by adapting an existing analysis for projection methods to our context of which a prominent characteristic lies in varying matrices $h_t(\mathbf{A})$ during iterations, and ended up with a comprehensively tight rate $O(\frac{1}{\max\{\Delta, \epsilon\}} \log \frac{n}{\epsilon})$. In addition, we discussed the equivalence and subtle differences between projection and search methods that give rise to new insights on the step-size schemes, and conducted experimental studies to verify the established theories. Particularly, the proposed adaptive learning rate scheme for the VR-PCA works far better than those proposed previously. One limitation of our analyses lies in the great difficulties of their extensions to the block setting where $\mathbf{X} \in \mathbb{R}^{n \times k}$ for $k \geq 1$ (this work corresponds to $k = 1$), because $\mathbf{X}_t^\top \mathbf{A} \mathbf{X}_t$ is no longer a scalar but a matrix and the key matrix function $h_t(\mathbf{A})$ then is not well-defined for the analysis. We consider overcoming the underlying difficulties in this setting as an important future work.

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
