# Supplementary Material: A Comprehensively Tight Analysis of Gradient Descent for PCA

**Zhiqiang Xu, Ping Li**

Cognitive Computing Lab
Baidu Research
No. 10 Xibeiwang East Road, Beijing 100193, China
10900 NE 8th St. Bellevue, Washington 98004, USA
{xuzhiqiang04,liping11}@baidu.com

**Theorem 2** The Riemannian gradient descent for Problem (1) in the main text with step-size $\eta = O(1) \leq \frac{1}{\lambda_1 - \lambda_n}$ converges in $T = O(\frac{1}{\epsilon} \log \frac{n}{\epsilon})$ iterations, i.e., $\lambda_1 - \mathbf{x}_T^\top \mathbf{A} \mathbf{x}_T < \epsilon$.

**Proof** We assume again that $\lambda_1 > \lambda_n$, and $\eta \leq \frac{1}{\lambda_1 - \lambda_n}$ such that $h_t(\lambda_i) = 1 + \eta(\lambda_i - \mathbf{x}_t^\top \mathbf{A} \mathbf{x}_t) \geq 0$ for all $i$ and $t$. In what follows, we show that no matter whether $\lambda_1$ is significantly larger than $\lambda_2$ in the sense that $h_0(\lambda_1) \geq (1 + \frac{\delta}{2}) h_0(\lambda_2)$ for $0 < \delta \leq 2$, it always holds that $\lambda_1 - \mathbf{x}_T^\top \mathbf{A} \mathbf{x}_T < \frac{2}{\eta} \epsilon$. Throughout the proof, we take $T = \lceil \frac{2}{\delta} \log \frac{n(1+\tan^2 \theta_0)}{\epsilon} \rceil + 1$, where $\theta_0 = \theta(\mathbf{x}_0, \mathbf{v}_1)$.

$\boxed{\text{Case 1}}$ that $h_0(\lambda_1) \geq (1 + \frac{\delta}{2}) h_0(\lambda_2)$. Consider the polynomial

$$p_T(x) = \sqrt{(1 + \frac{\delta}{2}) h_0(\lambda_2)} \prod_{t=0}^{T-1} \frac{h_t(x)}{(1 + \frac{\delta}{2}) h_t(\lambda_2)}$$

and its matrix form $p_T(\mathbf{A}) = \sum_{i=1}^n p_T(\lambda_i) \mathbf{v}_i \mathbf{v}_i^\top = \mathbf{V}_n p_T(\mathbf{\Sigma}_n) \mathbf{V}_n^\top$, where $p_T(\mathbf{\Sigma}_n) = \text{diag}(p_T(\lambda_1), \cdots, p_T(\lambda_n))$. Since $\eta \leq \frac{1}{\lambda_1 - \lambda_n}$, $h_t(x)$ for all $t$ and thus $p_T(x)$ are nonnegative for $x \in [\lambda_n, \lambda_1]$. Particularly, on the one hand,

**Fact 1.** For $x \in [\lambda_n, \lambda_2]$, $h_0(\lambda_1) \geq (1 + \frac{\delta}{2}) h_0(x)$ implies that $h_t(\lambda_1) \geq (1 + \frac{\delta}{2}) h_t(x)$ for all $t$, by the following lemma

**Lemma 4** If $\eta \leq \frac{2}{\lambda_1 - \lambda_n}$ then $\mathbf{x}_{t+1}^\top \mathbf{A} \mathbf{x}_{t+1} \geq \mathbf{x}_t^\top \mathbf{A} \mathbf{x}_t$.

Thus, the first property of $p_T(x)$ is that

$$p_T(\lambda_1) = \sqrt{(1 + \frac{\delta}{2}) h_0(\lambda_2)} \frac{h_0(\lambda_1)}{(1 + \frac{\delta}{2}) h_0(\lambda_2)} \prod_{t=1}^{T-1} \frac{h_t(\lambda_1)}{(1 + \frac{\delta}{2}) h_t(\lambda_2)} \geq \sqrt{h_0(\lambda_1)}. \tag{6}$$

On the other hand, noting that $h_t(\lambda_2) \geq h_t(\lambda_i)$ for all $i \geq 2$, it's easy to see $p_T(x)$'s second property:

$$p_T(\lambda_i) \leq \sqrt{h_0(\lambda_2)} (1 + \frac{\delta}{2})^{-T + \frac{1}{2}}, \quad i = 2, \cdots, n. \tag{7}$$

We then can rewrite $\mathbf{x}_T$ from Eq. (4) in the main text as

$$\mathbf{x}_T = \frac{\prod_{t=0}^{T-1} (\mathbf{I} + \eta(\mathbf{A} - \mathbf{x}_t^\top \mathbf{A} \mathbf{x}_t \mathbf{I})) \mathbf{x}_0}{\| \prod_{t=0}^{T-1} (\mathbf{I} + \eta(\mathbf{A} - \mathbf{x}_t^\top \mathbf{A} \mathbf{x}_t \mathbf{I})) \mathbf{x}_0 \|_2} = \frac{\prod_{t=0}^{T-1} h_t(\mathbf{A}) \mathbf{x}_0}{\| \prod_{t=0}^{T-1} h_t(\mathbf{A}) \mathbf{x}_0 \|_2} = \frac{p_T(\mathbf{A}) \mathbf{x}_0}{\| p_T(\mathbf{A}) \mathbf{x}_0 \|_2}.$$

Let $[ \cdot ]_1$ be the best rank-1 approximation of a matrix for the Frobenius norm. For example, $[p_T(\mathbf{A})]_1 = p_T(\lambda_1) \mathbf{v}_1 \mathbf{v}_1^\top$, due to that $p_T(\lambda_1) \geq \sqrt{h_0(\lambda_1)} \geq \sqrt{h_0(\lambda_i)} \geq p_T(\lambda_i) \geq 0$ for all $i \geq 2$,

by Eq. (6)-(7). By Lemma 14 in Musco et al. [10], we have the following Frobenius-norm rank-1 approximation inequality:

$$\|p_T(\mathbf{A}) - \mathbf{x}_T\mathbf{x}_T^\top p_T(\mathbf{A})\|_F^2 \leq (1 + \tan^2\theta_0)\|p_T(\mathbf{A}) - [p_T(\mathbf{A})]_1\|_F^2. \tag{8}$$

For the remainder on the right, by Eq. (7), we have that

$$\|p_T(\mathbf{A}) - [p_T(\mathbf{A})]_1\|_F^2 = \|\sum_{i=2}^n p_T(\lambda_i)\mathbf{v}_i\mathbf{v}_i^\top\|_F^2 = \sum_{i=2}^n p_T^2(\lambda_i)$$

$$\leq (n-1)h_0(\lambda_2)(1+\tfrac{\delta}{2})^{-2T+1}, \tag{9}$$

where

$$\begin{aligned}
(1+\tfrac{\delta}{2})^{-2T+1} &< (1+\tfrac{\delta}{2})^{-2(T-1)} = \exp\{-2(T-1)\log(1+\tfrac{\delta}{2})\} \\
&\leq \exp\{-\tfrac{4}{\delta}\log\tfrac{n(1+\tan^2\theta_0)}{\epsilon}\tfrac{\delta/2}{1+\delta/2}\} \leq \tfrac{\epsilon}{n(1+\tan^2\theta_0)}.
\end{aligned} \tag{10}$$

For the rank-1 approximation error on the left, it holds that

$$\begin{aligned}
\|p_T(\mathbf{A}) - \mathbf{x}_T\mathbf{x}_T^\top p_T(\mathbf{A})\|_F^2 &= \|p_T(\mathbf{A})\|_F^2 - \|\mathbf{x}_T\mathbf{x}_T^\top p_T(\mathbf{A})\|_F^2 \\
&= \|p_T(\mathbf{\Sigma}_n)\|_F^2 - \|\mathbf{x}_T^\top\mathbf{V}_n p_T(\mathbf{\Sigma}_n)\|_F^2 \\
&= \sum_{i=1}^n (1 - (\mathbf{x}_T^\top\mathbf{v}_i)^2)p_T^2(\lambda_i) \geq (1-(\mathbf{x}_T^\top\mathbf{v}_1)^2)p_T^2(\lambda_1) \\
&\geq (1-(\mathbf{x}_T^\top\mathbf{v}_1)^2)h_0(\lambda_1),
\end{aligned} \tag{11}$$

where the second equality is due to the orthogonal invariance for the Frobenius norm. By Eq. (8)-(11), we then get that $(1 - (\mathbf{x}_T^\top\mathbf{v}_1)^2)h_0(\lambda_1) < \epsilon h_0(\lambda_2)$. Hence, it holds that

$$h_0(\lambda_1) - \mathbf{x}_T^\top h_0(\mathbf{A})\mathbf{x}_T = h_0(\lambda_1) - \sum_{i=1}^n (\mathbf{x}_T^\top\mathbf{v}_i)^2 h_0(\lambda_i) \leq (1 - (\mathbf{x}_T^\top\mathbf{v}_1)^2)h_0(\lambda_1) < \epsilon h_0(\lambda_2),$$

which gives us $\lambda_1 - \mathbf{x}_T^\top\mathbf{A}\mathbf{x}_T < \tfrac{2}{\eta}\epsilon$, by noting that $h_0(\lambda_1) - \mathbf{x}_T^\top h_0(\mathbf{A})\mathbf{x}_T = \eta(\lambda_1 - \mathbf{x}_T^\top\mathbf{A}\mathbf{x}_T)$ and $h_0(\lambda_i) = 1 + \eta(\lambda_i - \mathbf{x}_0^\top\mathbf{A}\mathbf{x}_0) \leq 1 + \eta(\lambda_i - \lambda_n) \leq 2$ for all $i$.

$\boxed{\text{Case 2}}$ that $h_0(\lambda_1) < (1 + \tfrac{\delta}{2})h_0(\lambda_2)$. Consider the polynomial

$$q_T(x) = \sqrt{h_0(\lambda_1)}\prod_{t=0}^{T-1}\frac{h_t(x)}{h_t(\lambda_1)}.$$

We can write that

$$\mathbf{x}_T = \frac{\prod_{t=0}^{T-1} h_t(\mathbf{A})\mathbf{x}_0}{\|\prod_{t=0}^{T-1} h_t(\mathbf{A})\mathbf{x}_0\|_2} = \frac{q_T(\mathbf{A})\mathbf{x}_0}{\|q_T(\mathbf{A})\mathbf{x}_0\|_2}.$$

Define the index set

$$\alpha = \{i : \tfrac{1}{1+\frac{\delta}{2}}h_0(\lambda_1) \leq h_0(\lambda_i) < h_0(\lambda_1)\}.$$

Note that $|\alpha| \geq 1$ since $2 \in \alpha$. Let

$$\begin{aligned}
\mathbf{V}_\alpha &= \begin{bmatrix}\mathbf{v}_2 & \cdots & \mathbf{v}_{|\alpha|+1}\end{bmatrix}, & \mathbf{V}_{-\alpha} &= \begin{bmatrix}\mathbf{v}_1 & \mathbf{v}_{|\alpha|+2} & \cdots & \mathbf{v}_n\end{bmatrix}, \\
\mathbf{\Sigma}_\alpha &= \mathrm{diag}(\lambda_2, \cdots, \lambda_{|\alpha|+1}), & \mathbf{\Sigma}_{-\alpha} &= \mathrm{diag}(\lambda_1, \lambda_{|\alpha|+2}, \cdots, \lambda_n).
\end{aligned}$$

We then can have $q_T(\mathbf{A})$ decomposed into

$$\begin{aligned}
q_T(\mathbf{A}) &= \sum_{i\in\alpha} q_T(\lambda_i)\mathbf{v}_i\mathbf{v}_i^\top + \sum_{i\notin\alpha} q_T(\lambda_i)\mathbf{v}_i\mathbf{v}_i^\top \\
&= \mathbf{V}_\alpha q_T(\mathbf{\Sigma}_\alpha)\mathbf{V}_\alpha^\top + \mathbf{V}_{-\alpha} q_T(\mathbf{\Sigma}_{-\alpha})\mathbf{V}_{-\alpha}^\top \triangleq q_T(\mathbf{A}_\alpha) + q_T(\mathbf{A}_{-\alpha}),
\end{aligned}$$

and accordingly,

$$\mathbf{x}_T = \frac{q_T(\mathbf{A}_\alpha)\mathbf{x}_0}{\|q_T(\mathbf{A})\mathbf{x}_0\|_2} + \frac{q_T(\mathbf{A}_{-\alpha})\mathbf{x}_0}{\|q_T(\mathbf{A})\mathbf{x}_0\|_2} = \mathbf{V}_\alpha \frac{q_T(\mathbf{\Sigma}_\alpha)\mathbf{V}_\alpha^\top \mathbf{x}_0}{\|q_T(\mathbf{A})\mathbf{x}_0\|_2} + \mathbf{V}_{-\alpha} \frac{q_T(\mathbf{\Sigma}_{-\alpha})\mathbf{V}_{-\alpha}^\top \mathbf{x}_0}{\|q_T(\mathbf{A})\mathbf{x}_0\|_2}$$

$$\triangleq \mathbf{V}_\alpha \tilde{\mathbf{y}}_T^{(\alpha)} + \mathbf{V}_{-\alpha} \tilde{\mathbf{y}}_T^{(-\alpha)} \triangleq \tilde{\mathbf{x}}_T^{(\alpha)} + \tilde{\mathbf{x}}_T^{(-\alpha)} \triangleq \|\tilde{\mathbf{x}}_T^{(\alpha)}\|_2 \, \mathbf{x}_T^{(\alpha)} + \|\tilde{\mathbf{x}}_T^{(-\alpha)}\|_2 \, \mathbf{x}_T^{(-\alpha)}.$$

In order to analyze $\mathbf{x}_T^\top h_0(\mathbf{A})\mathbf{x}_T = \|h_0^{\frac{1}{2}}(\mathbf{A})\mathbf{x}_T\|_2^2$, we first check $\|h_0^{\frac{1}{2}}(\mathbf{A})\mathbf{x}_T^{(\alpha)}\|_2^2$ as follows:

$$\|h_0^{\frac{1}{2}}(\mathbf{A})\mathbf{x}_T^{(\alpha)}\|_2^2 = \frac{(\tilde{\mathbf{y}}_T^{(\alpha)})^\top \mathbf{V}_\alpha^\top h_0(\mathbf{A})\mathbf{V}_\alpha \tilde{\mathbf{y}}_T^{(\alpha)}}{\|\tilde{\mathbf{y}}_T^{(\alpha)}\|_2^2} = \frac{(\tilde{\mathbf{y}}_T^{(\alpha)})^\top h_0(\mathbf{\Sigma}_\alpha)\tilde{\mathbf{y}}_T^{(\alpha)}}{\|\tilde{\mathbf{y}}_T^{(\alpha)}\|_2^2}$$

$$\geq \frac{1}{1 + \frac{\delta}{2}} h_0(\lambda_1) \geq (1 - \frac{\delta}{2}) h_0(\lambda_1), \tag{12}$$

where the first inequality is by the definition of the index set. To check $\|h_0^{\frac{1}{2}}(\mathbf{A})\mathbf{x}_T^{(-\alpha)}\|_2^2$, similarly to Case 1, we consider the rank-1 approximation by $q_T(\mathbf{A}_{-\alpha})\mathbf{x}_0$. Note that $\mathbf{x}_T^{(-\alpha)} = \frac{\tilde{\mathbf{x}}_T^{(-\alpha)}}{\|\tilde{\mathbf{x}}_T^{(-\alpha)}\|_2} = \frac{q_T(\mathbf{A}_{-\alpha})\mathbf{x}_0}{\|q_T(\mathbf{A}_{-\alpha})\mathbf{x}_0\|_2}$. We then have the approximation inequality:

$$\|q_T(\mathbf{A}_{-\alpha}) - \mathbf{x}_T^{(-\alpha)}(\mathbf{x}_T^{(-\alpha)})^\top q_T(\mathbf{A}_{-\alpha})\|_F^2 \leq (1 + \tan^2 \theta_0)\|q_T(\mathbf{A}_{-\alpha}) - [q_T(\mathbf{A}_{-\alpha})]_1\|_F^2. \tag{13}$$

Here, noting $q_T(\lambda_1) = \sqrt{h_0(\lambda_1)} \geq q_T(\lambda_i)$ for all $i \geq 2$, it holds that

$$\|q_T(\mathbf{A}_{-\alpha}) - \mathbf{x}_T^{(-\alpha)}(\mathbf{x}_T^{(-\alpha)})^\top q_T(\mathbf{A}_{-\alpha})\|_F^2$$

$$= \|q_T(\mathbf{\Sigma}_{-\alpha})\|_F^2 - \|(\mathbf{x}_T^{(-\alpha)})^\top \mathbf{V}_{-\alpha} q_T(\mathbf{\Sigma}_{-\alpha})\|_F^2 = \sum_{i \notin \alpha}(1 - ((\mathbf{x}_T^{(-\alpha)})^\top \mathbf{v}_i)^2)q_T^2(\lambda_i)$$

$$\geq (1 - ((\mathbf{x}_T^{(-\alpha)})^\top \mathbf{v}_1)^2)q_T^2(\lambda_1) \geq (1 - ((\mathbf{x}_T^{(-\alpha)})^\top \mathbf{v}_1)^2)h_0(\lambda_1). \tag{14}$$

At the same time, since it holds at $t = 0$ by the definition of $\alpha$, then by Fact 1 we must have that $h_t(\lambda_i) \leq \frac{1}{1 + \frac{\delta}{2}} h_t(\lambda_1)$ for any $i \notin \{1\} \cup \alpha$. Thus, similarly to Eq. (9)-(10), we get that

$$\|q_T(\mathbf{A}_{-\alpha}) - [q_T(\mathbf{A}_{-\alpha})]_1\|_F^2 = \sum_{i \notin \{1\} \cup \alpha} q_T^2(\lambda_i) \leq (n - |\alpha| - 1)(1 + \frac{\delta}{2})^{-2T} h_0(\lambda_1) < \frac{h_0(\lambda_1)}{1 + \tan^2 \theta_0}\epsilon. \tag{15}$$

By Eq. (13)-(15), we can write that

$$h_0(\lambda_1) - (\mathbf{x}_T^{(-\alpha)})^\top h_0(\mathbf{A}_{-\alpha})\mathbf{x}_T^{(-\alpha)} = h_0(\lambda_1) - \sum_{i \notin \alpha}((\mathbf{x}_T^{(-\alpha)})^\top \mathbf{v}_i)^2)h_0(\lambda_i)$$

$$\leq h_0(\lambda_1) - ((\mathbf{x}_T^{(-\alpha)})^\top \mathbf{v}_1)^2)h_0(\lambda_1) < h_0(\lambda_1)\epsilon.$$

Thus, it holds that

$$\|h_0^{\frac{1}{2}}(\mathbf{A})\mathbf{x}_T^{(-\alpha)}\|_2^2 = \|h_0^{\frac{1}{2}}(\mathbf{A}_{-\alpha})\mathbf{x}_T^{(-\alpha)}\|_2^2 = (\mathbf{x}_T^{(-\alpha)})^\top h_0(\mathbf{A}_{-\alpha})\mathbf{x}_T^{(-\alpha)} > (1 - \epsilon)h_0(\lambda_1). \tag{16}$$

By Eq. (12) with $\delta = 2\epsilon$ (assuming $\epsilon \leq 1$) and Eq. (16), we get that

$$\mathbf{x}_T^\top h_0(\mathbf{A})\mathbf{x}_T = \|h_0^{\frac{1}{2}}(\mathbf{A})\mathbf{x}_T\|_2^2$$

$$= \|h_0^{\frac{1}{2}}(\mathbf{A})\tilde{\mathbf{x}}_T^{(\alpha)}\|_2^2 + \|h_0^{\frac{1}{2}}(\mathbf{A})\tilde{\mathbf{x}}_T^{(-\alpha)}\|_2^2$$

$$= \|\tilde{\mathbf{x}}_T^{(\alpha)}\|_2^2 \, \|h_0^{\frac{1}{2}}(\mathbf{A})\mathbf{x}_T^{(\alpha)}\|_2^2 + \|\tilde{\mathbf{x}}_T^{(-\alpha)}\|_2^2 \, \|h_0^{\frac{1}{2}}(\mathbf{A})\mathbf{x}_T^{(-\alpha)}\|_2^2$$

$$> (\|\tilde{\mathbf{x}}_T^{(\alpha)}\|_2^2 + \|\tilde{\mathbf{x}}_T^{(-\alpha)}\|_2^2)(1 - \epsilon)h_0(\lambda_1)$$

$$= (1 - \epsilon)h_0(\lambda_1),$$

and thus $h_0(\lambda_1) - \mathbf{x}_T^\top h_0(\mathbf{A})\mathbf{x}_T < \epsilon h_0(\lambda_1)$, i.e., $\lambda_1 - \mathbf{x}_T^\top \mathbf{A}\mathbf{x}_T < \frac{2}{\eta}\epsilon$.

Therefore, we have proved that $\lambda_1 - \mathbf{x}_T^\top \mathbf{A}\mathbf{x}_T < \frac{2}{\eta}\epsilon$ in both cases for $T = \lceil \frac{1}{\epsilon} \log \frac{n(1 + \tan^2 \theta_0)}{\epsilon} \rceil + 1$ (noting that we have taken $\delta = 2\epsilon$ in Case 2). Finally, as long as $\eta = O(1)$, we could write with $\epsilon$ rescaling that $\lambda_1 - \mathbf{x}_T^\top \mathbf{A}\mathbf{x}_T < \epsilon$ for $T = O(\frac{1}{\epsilon} \log \frac{n}{\epsilon})$. $\qquad \square$

We are left with proving Fact 1 and Lemma 4.

**Proof of Fact 1**  For any $t$ and $x \in [\lambda_n, \lambda_2]$,

$$h_t(\lambda_1) - (1 + \tfrac{\delta}{2})h_t(x)$$
$$= 1 + \eta(\lambda_1 - \mathbf{x}_t^\top \mathbf{A}\mathbf{x}_t) - (1 + \tfrac{\delta}{2})(1 + \eta(x - \mathbf{x}_t^\top \mathbf{A}\mathbf{x}_t))$$
$$= 1 + \eta(\lambda_1 - \mathbf{x}_0^\top \mathbf{A}\mathbf{x}_0) - (1 + \tfrac{\delta}{2})(1 + \eta(x - \mathbf{x}_0^\top \mathbf{A}\mathbf{x}_0))$$
$$\qquad + \eta(\mathbf{x}_0^\top \mathbf{A}\mathbf{x}_0 - \mathbf{x}_t^\top \mathbf{A}\mathbf{x}_t) - (1 + \tfrac{\delta}{2})\eta(\mathbf{x}_0^\top \mathbf{A}\mathbf{x}_0 - \mathbf{x}_t^\top \mathbf{A}\mathbf{x}_t)$$
$$= h_0(\lambda_1) - (1 + \tfrac{\delta}{2})h_0(x) - \tfrac{\delta}{2}\eta(\mathbf{x}_0^\top \mathbf{A}\mathbf{x}_0 - \mathbf{x}_t^\top \mathbf{A}\mathbf{x}_t) \geq 0,$$

where the last equality is by the hypothesis and Lemma 4. $\qquad\square$

**Proof of Lemma 4**  Let $\tilde{\mathbf{g}}_t = \widetilde{\nabla} f(\mathbf{x}_t)$. Then

$$\|\mathbf{x}_t - \eta \tilde{\mathbf{g}}_t\|_2^2 \, (\mathbf{x}_{t+1}^\top \mathbf{A}\mathbf{x}_{t+1} - \mathbf{x}_t^\top \mathbf{A}\mathbf{x}_t)$$
$$= (\mathbf{x}_t - \eta \tilde{\mathbf{g}}_t)^\top \mathbf{A}(\mathbf{x}_t - \eta \tilde{\mathbf{g}}_t) - \mathbf{x}_t^\top \mathbf{A}\mathbf{x}_t\|\mathbf{x}_t - \eta \tilde{\mathbf{g}}_t\|_2^2$$
$$= \mathbf{x}_t^\top \mathbf{A}\mathbf{x}_t - 2\eta \mathbf{x}_t^\top \mathbf{A}\tilde{\mathbf{g}}_t + \eta^2 \tilde{\mathbf{g}}_t^\top \mathbf{A}\tilde{\mathbf{g}}_t - (\,1 + \eta^2\|\tilde{\mathbf{g}}_t\|_2^2\,)\, \mathbf{x}_t^\top \mathbf{A}\mathbf{x}_t$$
$$= 2\eta \tilde{\mathbf{g}}_t^\top \tilde{\mathbf{g}}_t + \eta^2 \tilde{\mathbf{g}}_t^\top \mathbf{A}\tilde{\mathbf{g}}_t - \eta^2 \mathbf{x}_t^\top \mathbf{A}\mathbf{x}_t\|\tilde{\mathbf{g}}_t\|_2^2$$
$$= \eta \tilde{\mathbf{g}}_t^\top (2\mathbf{I} + \eta \mathbf{A} - \eta \mathbf{x}_t^\top \mathbf{A}\mathbf{x}_t\mathbf{I})\tilde{\mathbf{g}}_t$$
$$\geq \eta(2 + \eta(\lambda_n - \lambda_1))\|\tilde{\mathbf{g}}_t\|_2^2 = \eta(2 - \eta(\lambda_1 - \lambda_n))\|\tilde{\mathbf{g}}_t\|_2^2,$$

where we have used that $\mathbf{x}_t^\top \tilde{\mathbf{g}}_t = 0$ and

$$-\mathbf{x}_t^\top \mathbf{A}\tilde{\mathbf{g}}_t \;=\; -\mathbf{x}_t^\top \mathbf{A}(\mathbf{I} - \mathbf{x}_t\mathbf{x}_t^\top)\mathbf{A}\mathbf{x}_t$$
$$\;=\; -\mathbf{x}_t^\top \mathbf{A}(\mathbf{I} - \mathbf{x}_t\mathbf{x}_t^\top)^2\mathbf{A}\mathbf{x}_t = \tilde{\mathbf{g}}_t^\top \tilde{\mathbf{g}}_t.$$

Thus, when $2 - \eta(\lambda_1 - \lambda_n) \geq 0$, i.e., $\eta \leq \frac{2}{\lambda_1 - \lambda_n}$, it holds that $\mathbf{x}_{t+1}^\top \mathbf{A}\mathbf{x}_{t+1} \geq \mathbf{x}_t^\top \mathbf{A}\mathbf{x}_t$. $\qquad\square$