# OpenReview forum: "A Comprehensively Tight Analysis of Gradient Descent for PCA"
_NeurIPS.cc/2021/Conference — NeurIPS 2021 Poster_

### Official Review · Reviewer_g4Fd · 2021-07-01

**Rating:** 6
**Confidence:** 3

**Summary:**

In this paper, the authors considered Riemanninan gradient descent for solving the top eigenvector/value problem for real-symmetric matrices. Rates in two regimes are given, depending on the gap of the top eigenvalues compared to the desired accuracy. The analysis is based on polynomial recursion derived from the algorithm updates. Additionally, for positive-definite matrices, a stepsize choice is proposed based on the upper bound on the contraction factor as suggested by the recursion. Numerical experiments are conducted for the power iteration, projected GD and Riemanninan GD.

**Limitations And Societal Impact:**

Yes.

**Main Review:**

The relevant literature is adequately surveyed. The claims do look right but I think the writing can be somewhat improved. The analysis itself, I believe, is somewhat standard from the numerical analysis literature.

Regarding the result, I'm a bit puzzled by the mismatch of guarantee for Theorem 1 and 2 - one is in terms of eigenvector (iterate) and the other is in terms of eigenvalue (objective function). And related to this, wouldn't something like Wedin's theorem (or slight variation thereof) help translate the result on eigenvalue back to a guarantee for subspace recovery perhaps? It would be nice if the authors could help clarify. Both the statements for Theorem 1 and 2 are also tied to the stepsize that depends on the gap, it would make a much more convincing case if a rigorous argument can be made on how robust the convergence rate is w.r.t these unknown quantities.

I also have some doubts about the practicality of the suggested stepsize for large instances. The plots for the experiments are shown w.r.t \# iterations, but it would make much more sense to show the walltime instead.

Minor typos: At various points in the proof of Thm 1, the index for $\prod$ is reversed.

======== post author-response ==============

Thank you for addressing my comments -- I do think some of these points need better clarification in the paper.

**Time Spent Reviewing:**

2

---

> ### Author Response · Authors · 2021-08-09
> **Response to Reviewer g4Fd**
>
> We will try our best to improve the writing for better readability.
>
> 1) About the mismatch between Theorem 1 and Theorem 2. We aim to solve the optimization problem (1) in Line 32, Page 1. For an optimzation problem, it is common to state convergence results in terms of either the point distance of the final solution to optimal solutions or its objective distance to optimal solutions. Results of Theorem 1 are actually stated in terms of the point distance (see Lines 138-139 in Page 4), while those of Theorem 2 are stated in the objective distance (see Line 140 in Page 4). Moreover, Theorem 2 is about the worst-case result.
>
> 2) Wedin's theorem (which upper bounds perturbation of eigenspace in subspace distance by those of both the given matrix and corresponding gap) or variations such as invariant subspace sensitivity (e.g., Corollary 7.2.5 in book matrix computation) seem not applicable in our case. We guess the reviewer was wondering if there exists some kind of counterpart that can translate results between Theorem 1 and Theorem 2 just like that for minimizing L-smooth and $\mu$-strongly convex function $f(\mathbf{x})$ in $\mathbb{R}^{n}$:
> \\[
> \frac{\mu}{2}\\|\mathbf{x}-\mathbf{x}^{\star}\\|\_2\leq f(\mathbf{x})-f(\mathbf{x}^{\star}) \leq \frac{L}{2}\\|\mathbf{x}-\mathbf{x}^{\star}\\|\_2,
> \\]
> where $\mathbf{x}^{\star}$ represent the optimal solution of $\min f(\mathbf{x})$. In our case, $f(\mathbf{x})=-\frac{1}{2}\mathbf{x}^{\top}\mathbf{A}\mathbf{x}$ subject to $\\|\mathbf{x}\\|\_2=1$, which is smooth but nonconvex. We indeed have such a counterpart:
> \\[
> \frac{\Delta\_p}{2}\sin^{2}\theta(\mathbf{x}, \mathbf{V}\_p)\leq f(\mathbf{x})-f(\mathbf{v}\_1)=\frac{1}{2}(\lambda\_1 - \mathbf{x}^{\top}\mathbf{A}\mathbf{x}) \leq \frac{\lambda\_1 - \lambda\_n}{2} \sin^{2}\theta(\mathbf{x}, \mathbf{V}\_p)
> \\]
> for $\\|\mathbf{x}\\|\_2=1$ (see Lines 141-142 in Page 4 and Lemma 2 in [1]). However, the translated results may possibly be worse. For example, by the second inequality and Theorem 1, we can get $\lambda\_1 - \mathbf{x}_T^{\top}\mathbf{A}\mathbf{x}_T < \epsilon$ for $T=O(\frac{1}{\Delta}\log\frac{1}{\epsilon})$. But when $\Delta$ is much less than $\epsilon$, $T$ is clearly worse than the results in Theorem 2. On the other hand, by the first inequality and Theorem 2, we can get $\sin^{2}\theta(\mathbf{x}\_T, \mathbf{V}\_p)<\epsilon$ for $T=O(\frac{1}{\Delta_p\\,\epsilon}\log\frac{1}{\Delta_p\\,\epsilon})$, which is worse than the results in Theorem 1.
> Note above that $\Delta=\frac{\Delta\_p}{\lambda\_1-\lambda\_n}=\frac{\lambda\_p-\lambda\_{p+1}}{\lambda\_1-\lambda\_n}$ where $\Delta\_i=\lambda\_i - \lambda\_{i+1}$ and $\Delta\_p$ is the first nonzero gap.
>
> 3) The stepsizes in Theorems 1-2 don't really depend on the gap $\Delta\_{1}=\lambda_1-\lambda_2$ (suppose $\lambda_1 > \lambda_2$). Instead, it depends on the quantity $\lambda_1 - \lambda_n$ which is the sum of all the gaps: $\lambda_1 - \lambda_n = \sum_{i=1}^{n-1} (\lambda_{i}-\lambda_{i+1})$. Thus, in general, it has nothing to do with the commonly used gap $\Delta\_{1}$. Although $\lambda_1 - \lambda_n$ is unknow either, it is much easier to be roughly estimated by, e.g., matrix 1-norm. When the step-size is not greater than $\frac{1}{\lambda_1 - \lambda_n}$, it converges and larger stepsize results in faster convergence (see, e.g.,  the first equality in Line 128). Otherwise it may not converge.
>
> 4) Walltime is shown in Figures 2-3 but not in Figure 1. In particular, Figure 3 has clearly demonstrated the effectiveness of the proposed step-size compared to existing ones. We will follow your suggestion to report time comparison for Figure 1 (which is similar to the case of Figure 2).
>
> 5) The reverse index for $\prod$ follows the expansion of the recursion equation about the update from the last iteration T to the start 0. Since those matrices are commutative, the order of index does not matter. We will follow your suggestion to use a uniform order.
>
> [1] Xu, Z., Cao, X., & Gao, X. (2018, July). Convergence analysis of gradient descent for eigenvector computation. International Joint Conferences on Artificial Intelligence.

---

### Official Review · Reviewer_AfhS · 2021-07-13

**Rating:** 7
**Confidence:** 2

**Summary:**

In this work authors presented two analyses of the Riemannian gradient descent for eigenvector computation.  First authors provided a general tight analysis that holds for any real symmetric matrix. In addition, authors further give the first worst-case analysis that achieves a rate of convergence.

For real-world application, this gap-dependent analysis suggests a new promising learning rate for stochastic variance reduced PCA algorithms. Authors provided experiments to confirm the findings.

Overall this is a solid paper with both theoretical analysis and empirical studies.

**Ethical Concerns:**

Not observed

**Limitations And Societal Impact:**

I did not observe any potential negative societal impact of their work.


**Main Review:**

Originality: One suggestion is adding a table to compare the work with other related works on the assumptions and convergence rate (especially with Ding et al. 2020.). This would make it easier to compare and understand the contribution of this work.

Quality:  This paper has good quality with comprehensive theoretical analysis and rich empirical experiments.

Clarity: The writing is clear. The footnote in page 7 is a bit too long, authors may consider put it in the main text.
Also it is great to see the comparison w.r.t. both passes and wall-clock time (sec) in figure 2 and 3. I am curious if authors has any results for time (sec) comparison for figure 1?

Significance: This paper provides theoretical analysis for the Riemannian gradient descent for eigenvector computation. From the real world application, authors confirms it significantly improves the speed for VR-PCA.

**Time Spent Reviewing:**

2

---

> ### Author Response · Authors · 2021-08-09
> **Response to Reviewer AfhS**
>
> 1) Thank you very much for the good suggestion. We will add such a table for comparison.
>
> 2) We will put the footnote in Page 7 in the main text.
>
> 3) The time comparison for Figure 1 is similar to the case of Figure 2. We will follow your suggestion to add it.

---

### Official Review · Reviewer_ZjFy · 2021-07-16

**Rating:** 8
**Confidence:** 2

**Summary:**

This paper considers Riemannian gradient method for PCA. Contributions of the paper are following: 1) proved convergence rate of $O(\frac{1}{\max{\Delta, \epsilon}}\log\frac{1}{\varepsilon})$ for any real symmetric matrix; 2) showed without the need of the eigenvalue gap information that even if the gap $\Delta$ is significantly smaller than the target accuracy, it is possible to achieve a speed of convergence $O(\frac{1}{\epsilon}\log\frac{1}{\varepsilon})$. Numerical results verify obtained rate of convergence.

**Limitations And Societal Impact:**

Yes, the authors adequately addressed the limitations and potential negative societal impact of their work

**Main Review:**

Paper achieves new state-of-the-art theoretical results for Riemannian gradient method for PCA. Clear accept

**Time Spent Reviewing:**

4

---

> ### Author Response · Authors · 2021-08-09
> **Response to Reviewer ZjFy**
>
> Thank you very much for your comments.

---

### Official Review · Reviewer_UVrc · 2021-07-18

**Rating:** 6
**Confidence:** 3

**Summary:**

The paper studies Riemannian gradient descent for largest eigen-vector computation real symmetric matrices. For matrices with eigen-gap $\Delta$ and (global) convergence error $\epsilon$, the paper gives a gap-dependent convergence rate of $\frac{1}{\Delta} \log(1/\epsilon)$ and gap-independent analysis which gives a convergence rate of $\frac{1}{\epsilon} \log(1/\epsilon)$. This improves upon previous work that achieves a $\frac{1}{\Delta^2} \log(1/\epsilon)$ rate and generalizes related work that that achieves the same rate for projected gradient descent, power iteration and Riemannian gradient descent restricted to PSD matrices.

The paper highlights an omission in previous work that mistakenly stated a rate of $\frac{1}{\epsilon} \log(1/\epsilon)$ which is in fact $\frac{1}{\Delta^2\epsilon} \log(1/\epsilon)$.

Experiments comparing the power iteration, projected gradient descent and RGD are done for proposed adaptively chosen step-sizes.

**Ethics Review Area:**

["I don’t know"]

**Limitations And Societal Impact:**

Not applicable since the work is mainly theoretical.

**Main Review:**

Eigen-vector computation is a ubiquitous problem in machine learning and the improved convergence rate for RGD is an important contribution in understanding the convergence behaviour of PCA algorithms. The exploration of adaptive step-sizes in the experiments can also be a valuable line of investigation for future work.

That said, some concerns:
1. It seems like in Thm 1 and Thm 2 both have a dependence on $\log(\tan^2 \theta(x_0, V_p)))$. How do you bound this without any assumptions about the starting vector $x_0$? What am I missing here?
2. Line 49, Line 149, you mention that the analysis is adapted from [2] that uses "constant matrices". It isn't quite clear what this means. It seems like you mean the definition of $h_t(x)$ is different. A little more explanation may make the distinctino more clear.


[1] Xu, Z., Cao, X., & Gao, X. (2018, July). Convergence analysis of gradient descent for eigenvector computation. International Joint Conferences on Artificial Intelligence.

[2] Musco, C., & Musco, C. (2015). Randomized block krylov methods for stronger and faster approximate singular value decomposition. _arXiv preprint arXiv:1504.05477_.

**Time Spent Reviewing:**

3.5

---

> ### Author Response · Authors · 2021-08-09
> **Response to Reviewer UVrc**
>
> 1) Without any assumptions about $\mathbf{x}\_0$, we have $\tan^2\theta\in[0,\infty]$. Here $\tan^2\theta = 0$ means that $\mathbf{x}\_0$ is already in the column space of $\mathbf{V}\_p$ and thus $T=0$, while $\tan^2\theta = \infty$ means that $\mathbf{x}\_0$ is in the column space of the orthogonal complement of $\mathbf{V}\_p$ (denoted as $\mathbf{V}\_{-p}$) and thus the update steps can't make any progress toward $\mathbf{V}\_p$, i.e., $T=\infty$. Once $0<\tan^2\theta<\infty$ (i.e.,  an acute angle between $\mathbf{x}\_0$ and $\mathbf{V}\_p$), $\mathbf{x}\_t$ can step into the column space of $\mathbf{V}\_p$ as $t$ goes to $\infty$.
> A common practice is to use random $\mathbf{x}\_0$ which is entrywise standard Gaussian. Then we can bound $\tan^2\theta=\\|(\mathbf{V}\_{-p}^{\top}\mathbf{x}\_0)(\mathbf{V}\_{p}^{\top}\mathbf{x}\_0)^{\dagger}\\|\_{2}^{2}$ with high probability (w.h.p.). Note that both $\mathbf{V}\_{-p}^{\top}\mathbf{x}\_0$ and $\mathbf{V}\_{p}^{\top}\mathbf{x}\_0$ are also entrywise standard Gaussian since $\mathbf{V}\_{p}$ and $\mathbf{V}\_{-p}$ are orthonormal. By standard Gaussian matrix concentration theory (see, e.g., Mark Rudelson and Roman Vershynin, ICM 2010), w.h.p., $\\|\mathbf{V}\_{-p}^{\top}\mathbf{x}\_0\\|\_2^2\leq c\_1 n$ and $\|(\mathbf{V}\_{p}^{\top}\mathbf{x}\_0)^{\dagger}\|_{2}^{2}\leq c\_2 p$ for some fixed constants $c\_1, c\_2$. Thus, we have that $\tan^2\theta\leq cnp$ for a fixed constant $c$, w.h.p. Then $\log(1+\tan^2\theta)=O(\log(n))$ w.h.p. Assuming that $n=O(poly(\frac{1}{\epsilon}))$, we have that $\log(n(1+\tan^2\theta))=O(\log\frac{1}{\epsilon})$. We will add discussion on it.
>
> 2) In our analysis, we define $h\_t(x)$ to accommodate both constant and varying matrices during iterations. $h\_t(x)=x$ or $h\_t(\mathbf{A})=\mathbf{A}$ encodes the constant matrices which does not change with iteration $t$ in [2], while $h_t(x)=1+\eta(x-\mathbf{x}\_t^{\top}\mathbf{A}\mathbf{x}\_t)$ or $h_t(\mathbf{A})=\mathbf{I}+\eta(\mathbf{A}-\mathbf{x}\_t^{\top}\mathbf{A}\mathbf{x}\_t\cdot \mathbf{I})$ varies with iteration $t$ in our case. For analysis, constant matrices only need simple monomials in [2], while varying matrices require complicated polynomials that satisfy certain properties for us. We will make it clearer.

---

> > ### Comment · Reviewer_UVrc · 2021-09-03
> > **Response**
> >
> > Thank you for your detailed response. The $\log(n)$ dependence due to bounding the $\tan^2 \theta$ term is an important factor in my opinion that needs to be included.

---

> > > ### Author Response · Authors · 2021-09-04
> > > **Response to Reviewer UVrc**
> > >
> > > Thank you very much for your feedback.
> > >
> > > In general, it holds that $n=\mathrm{poly}(\frac{1}{\epsilon})$. We then can write $\log(n)=O(\log(\frac{1}{\epsilon}))$. Following our previous response, we thus can simplify the log term in the bound as follows:
> > > \\[
> > > \\begin{array}{l}
> > > \\quad\log\frac{n(1+\tan^2\theta)}{\epsilon}\newline
> > > =\log(n)+\log(1+\tan^2\theta))+\log\frac{1}{\epsilon}\\newline
> > > =\log(n)+O(\log(n))+\log\frac{1}{\epsilon}\quad(\textrm{using }\log(1+\tan^2\theta)=O(\log(n))\textrm{ w.h.p. from previous response})\newline
> > > =O(\log(n))+\log\frac{1}{\epsilon}\newline
> > > =O(\log\frac{1}{\epsilon})+\log\frac{1}{\epsilon}\quad(\text{using }n\textrm{'s representation: }n=\mathrm{poly}(\frac{1}{\epsilon})) \newline
> > > =O(\log\frac{1}{\epsilon})
> > > \\end{array}.
> > > \\]
> > > If we don't use $n$'s representation, i.e., $n=\mathrm{poly}(\frac{1}{\epsilon})$, we then can explicitly write that
> > > \\[
> > > \log\frac{n(1+\tan^2\theta)}{\epsilon}=O(\log\frac{n}{\epsilon}).
> > > \\]
> > >
> > > We will follow your suggestion to write in this explicit form of $n$.

---

### Official Review · Reviewer_KXps · 2021-08-02

**Rating:** 6
**Confidence:** 3

**Summary:**

This paper provides a tight analysis of riemannian gradient descent of its iteration complexity.

**Limitations And Societal Impact:**

I don't think this paper discuss limitations. Probably the author can comment on the case of computing a block of eigenvectors.

**Main Review:**

I think the paper is solid and a good work. It provides a gap free bound which is interesting. I am just wondering whether there would be any block version of the gradient descent method. If so, is it possible
to extend the analysis here to that setting. I notice the specific update form, e.g., (4), is crucial for the analysis. Can the author comment on this part.

**Time Spent Reviewing:**

3 hours

---

> ### Author Response · Authors · 2021-08-09
> **Response to Reviewer KXps**
>
> 1) In this work, we consider the vector setting for Problem (1), i.e., $\mathbf{x}\in\mathbb{R}^{n\times k}$  with $k=1$ and $\\|\mathbf{x}\\|\_{2}=1$, for which the update equation in Eq. (3) can be written as
> \begin{equation}
> \mathbf{x}\_{t+1}=\frac{\mathbf{x}\_t + \eta(\mathbf{I}-\mathbf{x}\_t \mathbf{x}\_t^{\top})\mathbf{A}\mathbf{x}\_t}{\\|\mathbf{x}\_t + \eta(\mathbf{I}-\mathbf{x}\_t \mathbf{x}\_t^{\top})\mathbf{A}\mathbf{x}\_t\\|_{2}}
> =\frac{(\mathbf{I}+\eta(\mathbf{A}-\mathbf{x}\_t^{\top}\mathbf{A}\mathbf{x}\_t\mathbf{I}))\mathbf{x}\_t}{\\|(\mathbf{I}+\eta(\mathbf{A}-\mathbf{x}\_t^{\top}\mathbf{A}\mathbf{x}\_t\mathbf{I}))\mathbf{x}\_t\\|_2}
> =\frac{h_t(\mathbf{A})\mathbf{x}\_t}{\\|h_t(\mathbf{A})\mathbf{x}\_t\\|_2}.
> \end{equation}
> Yes, the block setting of Problem (1) is
> $\displaystyle\min\_{\mathbf{X}\in\mathbb{R}^{n\times k}: \mathbf{X}^T\mathbf{X}=\mathbf{I}} f(\mathbf{X})=-\frac{1}{2}\mathrm{trace}(\mathbf{X}^{\top}\mathbf{A}\mathbf{X})$ with $k>1$. The update equation in Eq. (3) now
> becomes
> \begin{equation}
> \mathbf{X}\_{t+1}=(\mathbf{X}\_t + \eta(\mathbf{I}-\mathbf{X}\_t \mathbf{X}\_t^{\top})\mathbf{A}\mathbf{X}\_t)
> (\mathbf{I}+\eta^{2}\mathbf{X}^{\top}\mathbf{A}(\mathbf{I}-\mathbf{X}\_t \mathbf{X}\_t^{\top})\mathbf{A}\mathbf{X}\_t)^{-\frac{1}{2}}.
> \end{equation}
> However, it is quite difficult to extend the current analysis (which depends crucially on the second equality on the top about $\mathbf{x}\_{t+1}$) to the block setting, because $\mathbf{X}\_t^{\top}\mathbf{A}\mathbf{X}\_t$ is no longer a scalar but a matrix and thus the second equality on the top about $\mathbf{x}\_{t+1}$ does not hold any more.  Moreover, $h_t(\mathbf{A})$ becomes not well-defined in the block setting. We consider overcoming these difficulties of the extension to the block setting as an important future work.
>
> 2) Regarding Eq. (4), it is an analogue of expanding the power iteration:
> \\[
> \mathbf{x}\_{T}=\frac{\mathbf{A}\mathbf{x}\_{T-1}}{\\|\mathbf{A}\mathbf{x}\_{T-1}\\|\_2}=\frac{\mathbf{A}\times \frac{\mathbf{A}\mathbf{x}\_{T-2}}{\\|\mathbf{A}\mathbf{x}\_{T-2}\\|\_2}}{\\|\mathbf{A}\times \frac{\mathbf{A}\mathbf{x}\_{T-2}}{\\|\mathbf{A}\mathbf{x}\_{T-2}\\|\_2}\\|\_2}=\frac{\mathbf{A}^2\mathbf{x}\_{T-2}}{\\|\mathbf{A}^2\mathbf{x}\_{T-2}\\|\_2}=\cdots=\frac{\mathbf{A}^{T}\mathbf{x}\_{0}}{\\|\mathbf{A}^{T}\mathbf{x}\_{0}\\|\_2}.
> \\] Thus, it is amenable to an analysis similar to that of the power method. Moreover, it can be equivalently rewritten as the workable polynomial form (see Line 168) for a gap-free analysis.
>
> 3) We will follow your suggestion to discuss the limitations about the difficulties of extending current analysis to the block setting.

---

### Decision · Program_Chairs · 2021-09-27

**Decision:**

Accept (Poster)

**Comment:**

This paper studies an important problem and improves upon existing quantitative results. The reviewers agree that this is a solid contribution and in particular appreciate that both gap-dependent and gap-free bounds are obtained.

The reviewers also recognize as a major limitation of this work that the results only apply to finding the top eigenvectors. It is unclear how the current techniques can be generalized to top-k eigenvectors.

The paper can be made stronger if the authors can add (1) a table for quantitative comparison with existing work, (2) discussion of generalization to top-k, (3) discussion on the choice of initial vectors and the use of random initialization, and (4) discussion on practical implementation. These discussions will be useful to those who want to apply the results in this paper.